# LIPEx – Locally Interpretable Probabilistic Explanations – To Look Beyond The True Class

## Abstract

In this work, we instantiate a novel perturbation-based multi-class explanation framework, LIPEx (**L**ocally **I**nterpretable **P**robabilistic **Ex**planation). We demonstrate that LIPEx not only locally replicates the probability distributions output by the widely used complex classification models but also provides insight into how every feature deemed to be important affects the prediction probability for each of the possible classes. We achieve this by defining the explanation as a matrix obtained via regression in the space of probability distributions, with respect to the Hellinger distance. Ablation tests on text and image data, show that LIPEx-guided removal of important features from the data causes more change in predictions for the underlying model than similar tests based on other saliency-based or feature importance-based Explainable AI (XAI) methods. It is also shown that compared to LIME, LIPEx is more data efficient in terms of using a lesser number of perturbations of the data to obtain a reliable explanation. This data-efficiency is seen to manifest as LIPEx being able to compute its explanation matrix ~ 53% faster than all-class LIME, for classification experiments with text data.

## 1 Introduction

Recent momentum in deep learning research has brought forth the importance of interpreting models with complex architectures. In a wide range of areas where neural nets have made a successful foray, the methods of Explainable AI (XAI) have also found an important use to help understand the functioning of these novel predictors - like in climate science (Labe & Barnes, 2021), for solving partial differential equation (Linial et al., 2023), in high-energy physics (Neubauer & Roy, 2022), information retrieval (Lyu & Anand, 2023), in legal A.I. (Collenette et al., 2023), etc. Most often, it has been observed that models with complex architectures give better accuracy compared to a simple model. So, the core achievement of XAI methods is to be able to give a highly accurate local replication of a complex predictor's behaviour by a simple model over humanly interpretable components of the data (Ribeiro et al., 2016). Towards achieving this, multiple different XAI methods have been proposed in the recent times, e.g., LIME (Ribeiro et al., 2016), SHAP (Lundberg & Lee, 2017), Decision-Set (Lakkaraju et al., 2016), Anchor (Ribeiro et al., 2018), Smooth-GRAD (Smilkov et al., 2017b), Poly-CAM (Englebert et al., 2022), Extremal Perturbrations (Fong et al., 2019), Saliency Maps (Simonyan et al., 2014), etc.

A major motivation for explainability is to debug a model in a classification setup (Casillas et al., 2003; Dapaah & Grabowski, 2016). Towards this, an end user is not only interested in understanding the explanation provided for the predicted class at a particular data point but is also concerned about the influence of different features for all the possible class likelihoods estimated by a classifier. For any particular data point, the full spectrum of feature influence on each possible class can help to understand how well the model has been trained to discriminate a particular class from the rest. However, existing explanation frameworks do not provide this required multi-class view of the classification. To this end, we propose an explainability framework that can explain a classifier's output prediction beyond the true class.

To obtain an explanation around an input instance, a local explanation algorithm like LIME (Ribeiro et al., 2016; Garreau & Luxburg, 2020) creates perturbations of the data – each perturbation being represented as a Boolean vector. LIME includes a feature selection method to decide a set of important features for each class (like Algorithm A) among which the explanation is sought. The Boolean representation mentioned

previously is in as many dimensions as the number of these important features. The LIME explanation vector for the complex model's prediction on the input instance is obtained by solving a penalized linear regression over these Boolean represented perturbations labelled by the complex classifier's prediction for each of these perturbations. We posit that it is not entirely convincing that LIME attempts to regress over bounded labels i.e. probabilities, using an unbounded function (i.e. a linear function) and that this needs to be called separately for each class. Further, even if repeated calls are made on each possible class, there is no guarantee that by these repeated evaluations, the importance of any particular feature would be knowable for every class.

In this work, we attempt to remedy these problems by proposing an unified framework that can be applied to both text and image data. In a $\mathcal{C}$−class classification task, for any data $\boldsymbol{x}$ which is represented as $\boldsymbol{z}$ in some $f_{\boldsymbol{x}}$ dimensional feature space, we shall seek explanations that map into $\mathcal{C}$−class probability space as,

$$\mathbb{R}^{f_{\boldsymbol{x}}} \ni \boldsymbol{z} \mapsto \mathrm{Soft-Max} \circ \boldsymbol{Wz} \tag{1}$$

We will name the matrix $\boldsymbol{W} \in \mathbb{R}^{\mathcal{C} \times f_{\boldsymbol{x}}}$ as the "explanation matrix" – which can be obtained by minimizing some valid distance function (like Hellinger's distance) between distributions obtained as above and the probability distribution over classes that the complex model has been trained to map any input. Thus, we instantiate this novel way to implement perturbation based XAI, namely LIPEx.

Note that the matrix $\boldsymbol{W}$ in Equation 1 simultaneously gives for every feature a numerical measure of its importance for each possible class. We posit that it is important that in any explanation being done one a very good complex model, it should be evident that most features deemed to be important for the predicted class are not so for the other classes - an idea that was recently formalized in Gupta & Arora (2020); Gupta et al. (2022) for the specific case of saliency maps. In our method, this property turns out to be emergent as a consequence of the more principled definition of explanation that we start from.

Figure 1 shows an example of our matrix explanation obtained for a text document. We observe how the explanation matrix gives the contributions of a set of feature words to each possible class. Note that for the first row (the top predicted class), the top 5 feature words detected for this instance ([feel, valued, joy, treasures, incredibly]) are *distinctly different* from the top features detected for the class in the second row, the one with the second highest probability predicted by the classifier. And it is observed that for LIPEx there always arises a natural discrimination between features important for the different classes. More examples can be found in Appendix D.2 where we show qualitative comparisons between outputs from LIPEx and multi-class LIME. Similarly Figure 2 shows a comparison between results of different saliency methods and LIME and the LIPEx explanation on five image samples with true labels being (English springer, golf ball, French horn, church and garbage truck) respectively from the top.

In the following, we summarize our contributions towards formalizing this idea of matrix-based explanations.

**Novel Way To Do Perturbation Based XAI - That Explains The Distribution of A Classifier** In Section 3, we formally state our explanation approach, which can extract the relative importance of a set of features for every potential class under consideration. In Appendix D.4 we give a check that this method gets closer to human intuition than LIME when both can be compared on examples of human annotations.

In the following quantitative tests, we demonstrate the performance of LIPEx compared to existing state-of-the-art XAI methods. *In the first two tests,* we focus on LIPEx's ability to reproduce the output distribution of the complex model. *In tests three, four and five,* we focus on the importance of the features detected by LIPEx and its significant computational advantages over LIME in terms of run-time.

**(Test 1) Evidence of LIPEx Replicating the Complex Model's Predicted Distribution Over Classes** In Figure 3, we display the statistics over multiple data of the Total Variation (TV) distance between the output distribution of the obtained LIPEx surrogate explainer and that of the complex model on the same test instance. The results show that over hundreds of randomly chosen test instances, the distance is overwhelmingly near 0. We demonstrate that this necessary property of LIPEx holds over multiple models over text as well as images.

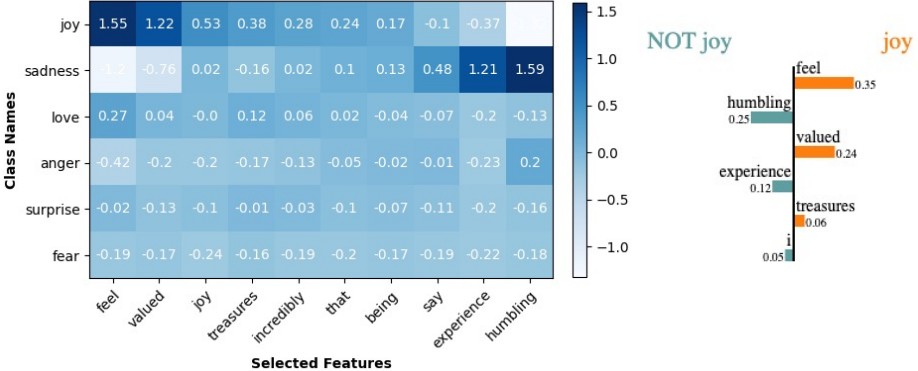

**Text**: *I have the joy of allowing kids to feel like the valued treasures that they are and to just have a blast being a kid alongside with them but can i just say its an incredibly humbling experience to have influence into a childs life and to know that what you do and say is being internalized.*

Figure 1: Example of comparison of explanation matrix obtained by LIPEx and the bar chart obtained by LIME on a text data from the Emotion dataset. For the LIPEx matrix, the class names on the left side are arranged in descending order of the predicted class probabilities. While the value of the numbers in the matrix represents the contribution of the selected features to the corresponding class.

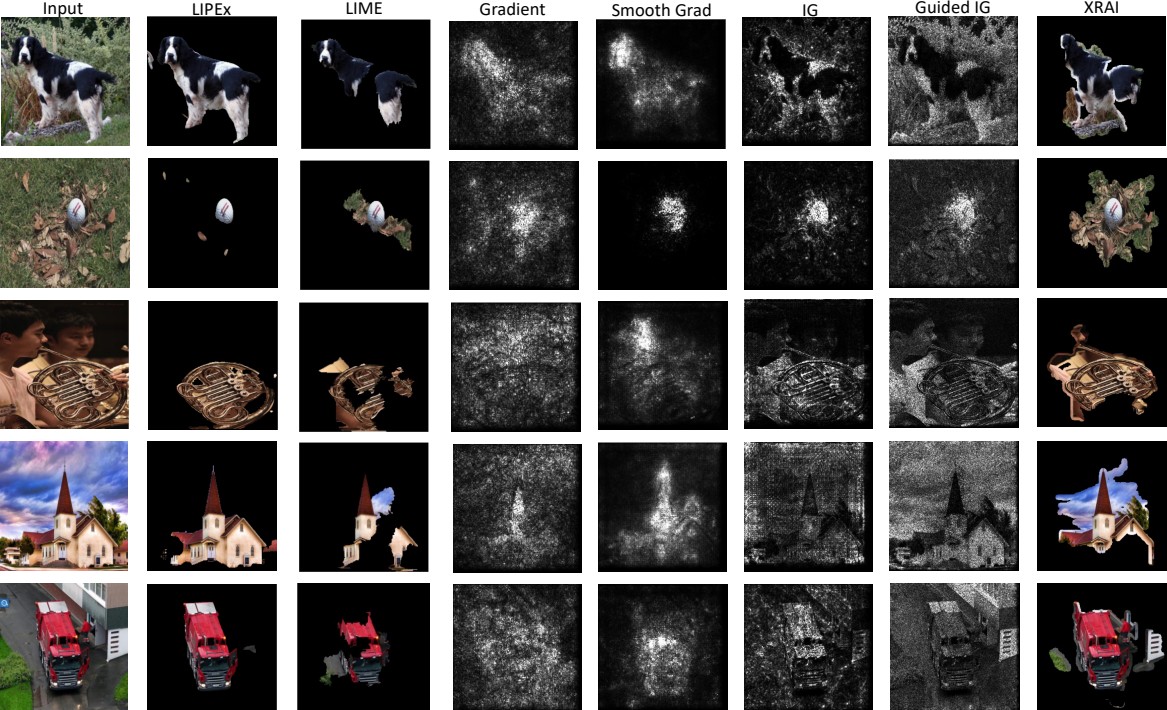

Figure 2: Example of Image explanation generated by LIPEx and other baseline methods on top predicted class by VGG16. Here, we default set Top-6 features for LIPEx explanation visualization.(LIME uses default Top-5 features for visualization and XRAI shows Top-30% most salient area ). All the test images are randomly chosen from the Imagenette dataset and the predicted class is the same as the true class.

**(Test 2) Sanity Check of LIPEx's Sensitivity to Model Distortion** In Figure 4 we distort a well-trained complex model by adding mean zero noise to the last layer parameters and measure how upon increasing the noise variance, the output probability distribution moves away in TV distance from the originally predicted probability distribution. We show that, averaged over test data, our surrogate model's predicted distribution evaluated over the distorted models, moves away from its original value almost identically. This sanity check is inspired by the arguments in Adebayo et al. (2018).

The above two tests give robust evidence that, indeed LIPEx is an accurate local approximator of the complex model while being dramatically simpler than the black-box predictor. To the best of our knowledge, such a strong model replication property is not known to be true for even the saliency methods, which can in-principle be called on different classes separately to get the relative importance among the different pixels for each class - albeit separately.

**(Test 3) Evidence of Changes in the Complex Model's Prediction Under LIPEx Guided Data Distortion** In Table 1 and 2, we demonstrate an ablation study guided by the "faithfulness" criteria as outlined in Atanasova et al. (2020). We establish that the top features detected by LIPEx are more important for the complex model than those detected by other XAI methods. We show this by demonstrating that upon removal of the top features from the original data and inference being done on this damaged data, the new predicted top class differs from the original top prediction for much more fraction of data when the removal is guided by LIPEx than other XAI methods.

**(Test 4) Evidence of LIPEx Replicating the Complex Model's Class Prediction Under LIPEx Guided Data Distortion** In Table 3, we demonstrate that for an overwhelming majority of data, upon removing their features deemed important by LIPEx, the new class predicted by the complex predictor is reproduced by the LIPEx map when its input the same distorted data.

**(Test 5) Stability of LIPEx While Using Few and Near-Input Perturbations of the Data - and the Consequential Speed-Up Over LIME** In Figure 5, we have verified that the features picked out by the LIPEx are largely stable when using only a few perturbations near the true data. We also show that this stability does not hold for LIME and thus LIPEx is demonstrably more data efficient.

To see the above in context, we recall that estimates were given in Agarwal et al. (2021) for the number of perturbations of the true data that are sufficient for LIME to produce reliable results - and this experiment of ours can be seen to corroborate that. Also, we recall that in works like Slack et al. (2020) it was pointed out that LIME's reliance on perturbations far from the true data creates a vulnerability that can be exploited to create adversarial attacks.

In Section 4.1 we present evidence that the data efficiency of LIPEx over LIME vividly shows up as significantly less computation time being required to compute our explanation matrix as opposed to the time required to generate the LIME explanation for all the classes.

Note that we have restricted our attention to "intrinsic evaluations" of explanations, i.e., we only use calls to the model as a black-box for deciding whether the explanations obtained are meaningful as opposed to looking for external human evaluation. Both text and image data were used to evaluate our proposed approach. For text-based experiments we used 20Newsgroup [1] and Emotion [2] datasets. For image-based experiments, we have used the Imagenette[3] dataset with segments detected by "segment anything" [4].

Among the above experiments, LIPEx was compared against a wide range of explanation methods, i.e., LIME (Ribeiro et al., 2016), Guided Backpropagation (Springenberg et al., 2014), Vanilla Gradients (Erhan et al., 2009), Integrated Gradients (Sundararajan et al., 2017), Deeplift (Shrikumar et al., 2016), Occlusion (Zeiler & Fergus, 2014), XRAI (Kapishnikov et al., 2019), GradCAM (Selvaraju et al., 2017), GuidedIG (Kapishnikov et al., 2021) and SmoothGrad (Smilkov et al., 2017a).

---

[1] http://qwone.com/~jason/20Newsgroups/
[2] https://huggingface.co/datasets/dair-ai/emotion
[3] https://github.com/fastai/imagenette
[4] https://segment-anything.com/

**Organization** In Section 2 we briefly overview related works in XAI. In Section 3 we give the precise loss function formalism for obtaining our explanation matrix, and in Section 4 all the tests will be given - comparing the relative benefits over other XAI methods. We conclude in Section 5. Appendices contain details such as the precise pseudocode used in Section 4 (in Appendix C), the hyperparameter settings (in Appendix B), and further experimental data is given in Appendix D.

## 2 Related Work

The work in Letham et al. (2015) is one of the first works that attempted to develop a classifier using rules and Bayesian analysis. In Ribeiro et al. (2016) a first attempt was made to describe explainability formally. The explanation can be made through an external explainer module, or a model can also be attempted to be made inherently explainable (Chattopadhyay et al., 2022). Post-hoc explainer strategy, as is the focus here, can be of different types, like (a) Ribeiro et al. (2016); Lundberg & Lee (2017) estimate feature importance for predicting a particular output, (b) counterfactual explanations (Wachter et al. (2017); Ustun et al. (2019); Rawal & Lakkaraju (2020)) (c) contrastive approaches (Jacovi et al., 2021) or (d) Weinberger et al. (2023) and Crabbé & van der Schaar (2022) where new XAI methods have been proposed tuned to the case of unsupervised learning. In this work, we specifically focus on feature importance-based explanation techniques.

**Feature Importance-based Explanations** The study in Ribeiro et al. (2016) initiated the LIME framework which we reviewed in Section 1 as our primary point of motivation. Similarly, the work in Lundberg & Lee (2017) used a statistical sampling approach ("SHAP") to explain a classifier model in terms of human interpretable features. Lakkaraju et al. (2016) proposed a decision set-based approach to train a classifier that can be interpretable and accurate simultaneously - where a set of independent if-then rules defines a decision set. Ribeiro et al. (2018) proposed an anchor-based approach for explanation - where anchors were defined as a set of sufficient conditions for a particular local prediction.

Evaluation is a critical component in any explanation framework. Doshi-Velez & Kim (2017) described important characteristics for the evaluation of explanation approaches. Evaluation criteria for explanations can broadly be categorized into two types, (a) criteria which measure how well the explainer module is able to mimic the original classifier and (b) criteria which measure the trustworthiness of the features provided by the explainer module, like the work in Qi et al. (2019) demonstrated the change in the prediction probability of a classifier with the removal of top $K$ features predicted by a saliency map explainer. Note, our tests done in Section 4 encompass both the above kinds of criteria.

Researchers have also delved into contrastive explanations Yin & Neubig (2022); Dhurandhar et al. (2018) and counterfactual explanations Wachter et al. (2017); Ustun et al. (2019). Contrastive explanations focus on explaining why a model predicted a certain class and not another. In counterfactual explanations, the question is of identifying the changes needed in the input to shift the model's prediction from one class to another. Hence counterfactual or contrastive explanations always deal with only two classes at a time and cannot be used to determine relative importance of any feature between all the possible classes. But our approach is inherently multi-class and is able to identify the importance of any feature for each of the classes.

Lastly, we note that in Sokol & Flach (2020) a tree-based explanation was attempted which could directly work in the multiclass setting but to be able to compete LIME their method's computation cost can need to scale with the number of features of an input data.

In summary, we note that in sharp contrast to our LIPEx proposal, none of the above existing methods have the critical ability to explain/reproduce the output distribution over classes that the given complex model would have produced.

## 3 Our Setup

Let $\mathcal{C} \in \{1, 2, 3, \ldots\}$ be the number of classes in the classification setup. Given any probability vector $\boldsymbol{p} \in [0,1]^{\mathcal{C}}$, $\sum_{i=1}^{\mathcal{C}} p_i = 1$, & $p_i \geq 0, \forall i$, we succinctly represent $\boldsymbol{p}$ as being a member of the simplex in $\mathcal{C}$–dimensions as $\boldsymbol{p} \in \Delta^{\mathcal{C}}$.

**Classifier Setup**  We aim to explain a classifier $h_{\boldsymbol{w}}$ (parameterized by weight $\boldsymbol{w}$), obtained by composing a neural network $\mathcal{N}$ and a Soft-Max layer, so that the output of the composition is a probability distribution over the $\mathcal{C}-$classes.

$$h_{\boldsymbol{w}} : \mathbb{R}^d \rightarrow \Delta^{\mathcal{C}}, \boldsymbol{x} \mapsto \text{SoftMax} \circ \mathcal{N}(\boldsymbol{x}) \tag{2}$$

In the above, a candidate $\boldsymbol{x} \in \mathbb{R}^d$ could be an original representation of an input instance (a sentence or an image). This composed function $h_{\boldsymbol{w}}$ in Equation 2 commonly is trained via the cross-entropy loss on a $\mathcal{C}$ class labelled data set and we assume only black-box access to it.

**Data Representation and Perturbation**  For a specific instance $\boldsymbol{x}$, we denote the number of unique features (tokens for text data or segments for image data) as $|\boldsymbol{x}|$, and we use an all-ones vector $\mathbf{1} \in \mathbb{R}^{|\boldsymbol{x}|}$ to represent the input instance $\boldsymbol{x}$, Hence, in such setups, the perturbations are generated by randomly dropping one or more features from original input instance $\boldsymbol{x}$ and these perturbations of the data $\boldsymbol{x}$ are denoted as binary vectors $\boldsymbol{z} \in \{0,1\}^{|\boldsymbol{x}|}$.

We assume that there is a pre-chosen function (say $T$) that maps the perturbed data points into the $d-$dimensional original data representation space which is the space of input instances to the original classifier $h_{\boldsymbol{w}}$ (Equation 2) .

$$T : \{0,1\}^{|\boldsymbol{x}|} \rightarrow \mathbb{R}^d \tag{3}$$

**The Feature Space for Explanations**  We define a feature selection function "Select" (like, Algorithm A), which maps,

$$\text{Select} : \{0,1\}^{|\boldsymbol{x}|} \rightarrow \{0,1\}^{f_{\boldsymbol{x}}}$$

where $f_{\boldsymbol{x}} < |\boldsymbol{x}|$ is the number of top important features to be selected for each perturbation of the input instance $\boldsymbol{x}$.

We define $\boldsymbol{z}$ as the Boolean space representation of the instance $\boldsymbol{x}$ and $\mathcal{S}(\boldsymbol{z}) \subseteq \{0,1\}^{|\boldsymbol{x}|}$ as the set of perturbations of $\boldsymbol{x}$. We define the corresponding set of perturbations after feature selection as,

$$\mathcal{S}(\boldsymbol{z}') := \text{Select}(\mathcal{S}(\boldsymbol{z})) \subseteq \{0,1\}^{f_{\boldsymbol{x}}} \subset \{0,1\}^{|\boldsymbol{x}|} \tag{4}$$

In above we have defined $\boldsymbol{z}' \in \{0,1\}^{f_{\boldsymbol{x}}}$ as the vector with only $f_{\boldsymbol{x}}$ features selected from $\boldsymbol{z}$.

Thus $\mathcal{S}(\boldsymbol{z}')$ is the feature space for our explanations.

**The Local Explanation Matrix**  We explain $h_{\boldsymbol{w}}$'s behaviour around $\boldsymbol{x}$ by a "simple surrogate model $g_{\boldsymbol{x},\boldsymbol{W}}$", which is constructed by composing a linear layer by a Soft-Max layer and is defined as,

$$g_{\boldsymbol{x},\boldsymbol{W}} : \mathcal{S}(\boldsymbol{z}') \rightarrow \Delta^{\mathcal{C}}, \ \boldsymbol{z}'' \mapsto \text{SoftMax} \circ \boldsymbol{W}\boldsymbol{z}'' \tag{5}$$

where the "explanation matrix" $\boldsymbol{W} \in \mathbb{R}^{\mathcal{C} \times f_{\boldsymbol{x}}}$. Note that the input dimensions $d$ of $h_{\boldsymbol{w}}$ and $f_{\boldsymbol{x}}$ of our explainer $g_{\boldsymbol{x},\boldsymbol{W}}$ could be very different and typically, $f_{\boldsymbol{x}} \lll d$.

Hellinger distance between two discrete distributions $\boldsymbol{p}, \boldsymbol{q}$ (on a set of $C$ possible classes) is given as,

$$\text{H}(\boldsymbol{p},\boldsymbol{q}) := \frac{1}{\sqrt{2}} \cdot \sqrt{\sum_{c \, \in \, \mathcal{C}} \left( \sqrt{\boldsymbol{p}(c)} - \sqrt{\boldsymbol{q}(c)} \right)^2}$$

Apart from being an intuitive symmetric measure, squared Hellinger distance (HD) also offers other attractive features of being sub-additive, smaller than half of the Kullback–Leibler divergence (KL) and always being within a quadratic factor of the Total Variation (TV) distance. So we chose to use the squared HD[5] to measure the distance between the output distribution of $h_{\boldsymbol{w}}$ and $g_{\boldsymbol{x},\boldsymbol{W}}$.

---

[5]We also tried TV but it underperformed compared to the squared Hellinger metric.

**The Explainer's Loss Function in the Boolean Perturbation Space** Let $\tilde{\mathcal{S}}(\boldsymbol{z}) \subset \mathcal{S}(\boldsymbol{z}) (\subseteq \{0,1\}^{|x|})$ be a randomly sampled set of data perturbations in feature space to be used for training. Passing it through the Select map (Equation 4) we obtain the feature space representations of the perturbations. We posit that the outputs of the Select map would determine what the explainer $g_{\boldsymbol{x},\boldsymbol{W}}$ in Equation 5 acts on. Further noting that the output of the $T$ in Equation 3 determines what the classifier $f_{\boldsymbol{w}}$ gets as input, we consider the following empirical risk function corresponding to a distance function $\pi$ in $\mathbb{R}^{|x|}$,

$$\hat{\mathcal{L}}_{\mathrm{H}}(g_{\boldsymbol{x},\boldsymbol{W}}, \tilde{\mathcal{S}}(\boldsymbol{z})) = \frac{1}{\left|\tilde{\mathcal{S}}(\boldsymbol{z})\right|} \sum_{\boldsymbol{y} \in \tilde{\mathcal{S}}(\boldsymbol{z})} \pi(\mathbf{1}, \boldsymbol{y}) \cdot \mathrm{H}^2\left(g_{\boldsymbol{x},\boldsymbol{W}} \circ \mathrm{Select}(\boldsymbol{y}), h_{\boldsymbol{w}} \circ T(\boldsymbol{y})\right) + \frac{\lambda}{2} \cdot \|\boldsymbol{W}\|_F^2 \qquad (6)$$

where $\mathbf{1}$ is the all-ones vector in $\mathbb{R}^{|x|}$, $\pi(\mathbf{1}, \boldsymbol{y}) = 1 - \frac{\boldsymbol{y}^\top \mathbf{1}}{\|\boldsymbol{y}\| \cdot \|\mathbf{1}\|}$. It is immediately interpretable that Equation 6 takes a $\pi-$weighted empirical average of the Hellinger distance squared between the true distribution over classes predicted by the complex classifier $h_{\boldsymbol{w}}$ and the distribution predicted by the explainer $g_{\boldsymbol{x},\boldsymbol{W}}$ while the $\lambda-$term penalizes for using high weight explainers and hence promotes simplicity of $g_{\boldsymbol{x},\boldsymbol{W}}$.

**Intuition for Good LIPEx Minima for Text Classifiers Being a ReLU Net** For intuition, consider explaining classification predictions by a ReLU net on a text data $\boldsymbol{x}$ with $|\boldsymbol{x}|$ unique words. Suppose the classifier has been trained to accept $d(\geq |\boldsymbol{x}|)$ word length texts in their embedding representation. Thus the $T$ map (Equation 3) that lifts the perturbations of $\boldsymbol{x}$ to the input space of the complex classifier can be imagined as a tall matrix of dimensions $d \times |\boldsymbol{x}|$ whose top $|\boldsymbol{x}| \times |\boldsymbol{x}|$ block is a diagonal matrix giving the embedding values for the words in this text and the rest of the matrix being zeros. Also, we note that the Select function can be imagined as a linear projection of the Boolean-represented perturbations into the subset of important features.

Further, any ReLU neural net is a continuous piecewise linear function . Hence, except at the measure zero set of non-differentiable points, the function $\mathcal{N}$ (Equation 2) is locally a linear function. Thus, for almost every input in $\mathbb{R}^d$ there exists a (possibly small) neighbourhood of it where $\mathcal{N} = \boldsymbol{W}_{\mathrm{net}}$ for some matrix $\boldsymbol{W}_{\mathrm{net}} \in \mathbb{R}^{\mathcal{C} \times d}$. It would be natural to expect that most true texts are not at the non-differentiable points of the net's domain and that $T$ maps small perturbations of the data into a small neighbourhood in the domain of the neural net. Hence, for many perturbations $\boldsymbol{y} \in \{0,1\}^{|\boldsymbol{x}|}$, $h_{\boldsymbol{w}}(T(\boldsymbol{y}))$, as it occurs in the loss in Equation 6, is a Soft-Max of a linear transformation (composition of the net and the $T$ map) of $\boldsymbol{y}$. Recall that this is exactly the functional form of the explainer $\boldsymbol{g}_{\boldsymbol{x},\boldsymbol{W}}$ (Equation 5) given that the Select function can be represented as a linear map! Thus, we see that there is a very definitive motivation for this loss function to yield good locally linear explanations for ReLU nets classifying text.

## 4 Results of the Evaluation Experiments

At the very outset, we note the following salient points about our setup. *Firstly,* that for any data $s$ (say a piece of text or an image), when implementing LIPEx on it, we generate a set of 1000 perturbations for each of the input instances. Then for each data, we chose its important features by taking a set union over the top-5 features of each possible class, which were returned by the "forward feature selection" method (reproduced in Algorithm A) called on the above set of data perturbations. We recall that this feature selection algorithm is standard in LIME implementations[6]. Suppose this union has $f_s$ features - then for all computations to follow for $s$ we always stick with these $f_s$ features for LIPEx (and also always call LIME on $f_s$ number of features in comparison experiments).

*Secondly,* we note that for the matrix returned by LIPEx (i.e. $\boldsymbol{W}$ in Equation 5) we shall define its "top$-k$" features as the features/columns of the matrix which give the $k-$highest entries by absolute value for the predicted class of that data.

**Reproducing the Distribution over Classes of the Complex Classifier** A key motivation of introducing the LIPEx was the need for the explanation methods to produce class distributions closely resembling those of the original classifier. Therefore, our initial emphasis is on investigating how much in Total Variation (TV) distance, the distribution over classes predicted by the obtained explainer is away from the one

---

[6]https://github.com/marcotcr/lime

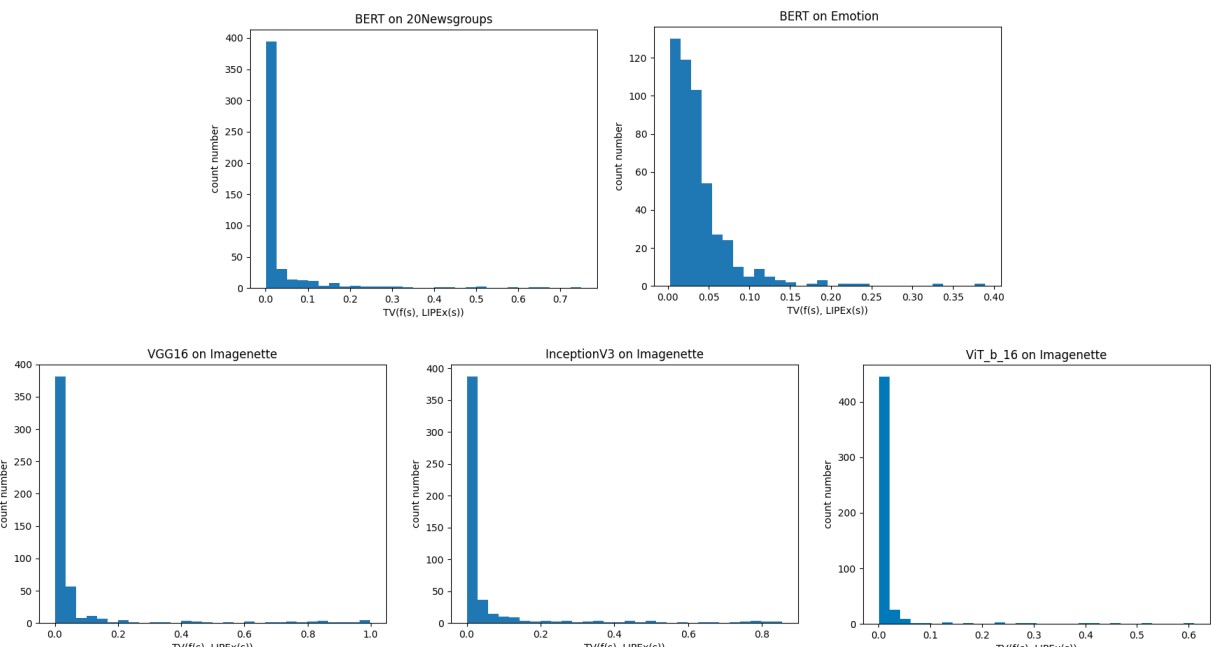

Figure 3: Histogram of the TV Distance between the probability distribution over classes as predicted by the complex classifier and that obtained by LIPEx.

predicted for the same data by the complex model. In Figure 3, we show the statistics of this TV distance for experiments on both text (i.e. BERT (Devlin et al., 2018) on 20Newsgroups and Emotion dataset) and image data (i.e. VGG16 (Simonyan & Zisserman, 2014), InceptionV3 (Szegedy et al., 2016), ViT_b_16 (Dosovitskiy et al., 2020) on Imagenette dataset).

Figure 3 clearly shows that the distribution is highly skewed towards 0 over 500 randomly sampled data.

Note that the LIPEx loss (Equation 6) never directly optimized for the TV gap to be small but the TV distance is upperbounded by $\sqrt{2}$ times the Hellinger distance between the distribution output by the LIPEx explainer and the complex model – which was optimized for. Hence we posit that this is a sanity check that LIPEx passes.

**LIPEx Tracks Distortions of The Complex Model's Output Distribution** This sanity check experiment is inspired by the studies in Adebayo et al. (2018). Here, mean-zero Gaussian noise is added to the trained complex model's last-layer weights and the noise variance is dialled up till the model's accuracy is heavily decreased. At each noise level, we compute the average Total Variation distance over randomly sampled data, between the output distribution of the damaged model and its original value, the same for the LIPEx's output distribution at respective inputs – and we also compute the average over data TV gap between the complex model and the LIPEx approximant of it. In Figure 4, We do text experiments with BERT on the Emotion dataset and image experiments with VGG16 and ViT on the Imagenette dataset.

In the figure legends, LIPEx is the LIPEx's output distribution for the original model, LIPEx($\sigma$) is the LIPEx's output distribution for the distorted model at noise variance $\sigma$. BERT, BERT($\sigma$), VGG16 and VGG16($\sigma$), ViT and ViT($\sigma$) are defined similarly.

The red curve in the figures on the bottom row of Figure 4 demonstrates that as the model distorts, LIPEx's output distribution remarkably closely (in TV distance) continues to track the distorted model's output distribution.

**Importance of Top-K Features Detected by LIPEx** A test of the correctness of determining any set of features to be important by an explanation method is that upon their removal from the original data

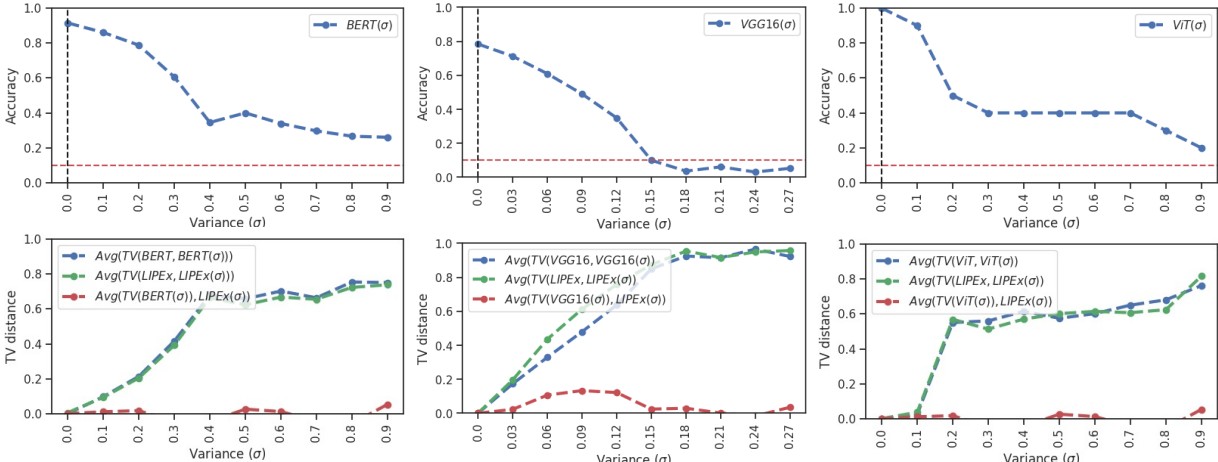

Figure 4: On the upper row, we see how the model accuracy drops upon adding noise with different variances. In the below row, we see how LIPEx tracks the changes in the complex model's output.

| Setting | Top-K | LIPEx | LIME | GuidedBack | Saliency | InputGrad | Deeplift | Occlusion |
|---------|-------|-------|------|-----------|----------|-----------|----------|-----------|
| BERT 20NewsGroups | K=1 | **0.78**(±0.05) | 0.78(±0.03) | 0.39(±0.05) | 0.39(±0.05) | 0.38(±0.05) | 0.38(±0.05) | 0.45(±0.05) |
| | K=2 | **0.86**(±0.04) | 0.84(±0.03) | 0.48(±0.08) | 0.48(±0.08) | 0.48(±0.08) | 0.48(±0.08) | 0.54(±0.07) |
| | K=3 | **0.90**(±0.02) | 0.86(±0.05) | 0.52(±0.07) | 0.52(±0.07) | 0.52(±0.07) | 0.52(±0.07) | 0.60(±0.08) |
| | K=4 | **0.91**(±0.03) | 0.88(±0.01) | 0.52(±0.05) | 0.52(±0.05) | 0.52(±0.05) | 0.52(±0.05) | 0.63(±0.07) |
| | K=5 | **0.92**(±0.03) | 0.90(±0.02) | 0.55(±0.04) | 0.55(±0.04) | 0.57(±0.03) | 0.57(±0.03) | 0.65(±0.06) |
| BERT Emotion | K=1 | **0.66**(±0.02) | 0.64(±0.02) | 0.60(±0.03) | 0.60(±0.03) | 0.60(±0.03) | 0.60(±0.03) | 0.65(±0.02) |
| | K=2 | **0.72**(±0.04) | 0.69(±0.04) | 0.61(±0.03) | 0.61(±0.03) | 0.62(±0.03) | 0.63(±0.03) | 0.66(±0.02) |
| | K=3 | **0.73**(±0.03) | 0.71(±0.04) | 0.64(±0.01) | 0.64(±0.01) | 0.64(±0.01) | 0.65(±0.01) | 0.67(±0.01) |
| | K=4 | **0.73**(±0.03) | 0.70(±0.05) | 0.62(±0.01) | 0.62(±0.01) | 0.63(±0.01) | 0.64(±0.01) | 0.70(±0.01) |
| | K=5 | **0.80**(±0.02) | 0.75(±0.01) | 0.63(±0.01) | 0.63(±0.01) | 0.64(±0.01) | 0.64(±0.01) | 0.69(±0.02) |

Table 1: Here, features refer to words. It can be seen that the features removed by LIPEx guidance more significantly impact the classifier's prediction than when guided by the other XAI methods.

and on presenting this modified/damaged input to the complex classification model it should produce a new predicted class than originally. We implement this experiment with text data in Table 1 and image data in Table 2. For saliency-based methods, the set of image segments is deduced by the Segment Anything[7] method and each segment's importance is determined as the sum of the weights assigned to its pixels by the saliency method. In each experiment 100 randomly sampled data points are used, and the tables give the mean and the standard deviation over three rounds of experiments.

Results demonstrate that when the top features detected by LIPEx are removed from the data, the original model's re-predicted class changes substantially more than when the same is measured for many other XAI methods, and the amount of change is proportional to the number of top features removed. [8]

**LIPEx Reproduces the Complex Model's Predictions Under LIPEx Guided Data distortion** We posit that for a multi-class explainer as LIPEx, it is a very desirable sanity check that it should reproduce the underlying model's (new) predicted classes on the input when its top features are removed. In Table 3, we run this experiment over 100 random sample text and image data, results show the re-prediction matching holds for the LIPEx for an overwhelming majority of data.

**Evidence for Data Efficiency of LIPEx as Compared to LIME** Since LIPEx and LIME, both are perturbation-based methods, a natural question arises if LIPEx is more data-efficient, or in other words can

---

[7]https://segment-anything.com/

[8]We use the code by Atanasova et al. (2020) to implement the gradient-based methods in Table 1, and the package https://github.com/PAIR-code/saliency to implement the saliency methods in Table 2.

| Setting | Top-K | LIPEx | LIME | VanillaGrad | SmoothGrad | IG | GuidedIG | XRAI |
|---|---|---|---|---|---|---|---|---|
| VGG16 Imagenette | K= 3 | **0.76**(±0.03) | 0.74(±0.01) | 0.68(±0.02) | 0.75(±0.02) | 0.70(±0.02) | 0.72(±0.03) | 0.71(±0.03) |
| | K= 4 | **0.82**(±0.01) | 0.78(±0.00) | 0.75(±0.01) | 0.82(±0.01) | 0.75(±0.03) | 0.75(±0.02) | 0.77(±0.01) |
| | K= 5 | **0.87**(±0.02) | 0.80(±0.01) | 0.81(±0.01) | 0.84(±0.01) | 0.77(±0.02) | 0.79(±0.01) | 0.80(±0.03) |
| InceptionV3 Imagenette | K= 3 | 0.67(±0.01) | 0.63(±0.01) | 0.66(±0.01) | 0.65(±0.02) | 0.64(±0.02) | 0.66(±0.05) | **0.69**(±0.02) |
| | K= 4 | **0.75**(±0.01) | 0.71(±0.05) | 0.65(±0.03) | 0.68(±0.05) | 0.71(±0.03) | 0.67(±0.05) | 0.70(±0.03) |
| | K= 5 | **0.77**(±0.02) | 0.77(±0.03) | 0.72(±0.05) | 0.74(±0.04) | 0.73(±0.01) | 0.71(±0.04) | 0.74(±0.06) |
| ViT_b_16 Imagenette | K= 3 | **0.32**(±0.05) | 0.30(±0.05) | 0.19(±0.04) | 0.22(±0.03) | 0.17(±0.02) | 0.18(±0.02) | 0.21(±0.04) |
| | K= 4 | **0.40**(±0.04) | 0.39(±0.05) | 0.24(±0.03) | 0.26(±0.04) | 0.24(±0.03) | 0.24(±0.02) | 0.26(±0.04) |
| | K= 5 | **0.46**(±0.03) | 0.45(±0.06) | 0.31(±0.05) | 0.34(±0.04) | 0.31(±0.03) | 0.28(±0.02) | 0.33(±0.04) |

Table 2: Here, features refer to image segments. In the above, we see that the fraction of data on which label prediction changes under deletion of top features detected by LIPEx is higher than for other methods.

| Setting | Modality | Top1 | Top2 | Top3 | Top4 | Top5 |
|---|---|---|---|---|---|---|
| BERT (20Newsgroups) | Text | 0.90 (±0.04) | 0.85(±0.02) | 0.79(±0.04) | 0.71(±0.03) | 0.70(±0.01) |
| BERT (Emotion) | Text | 0.89(±0.02) | 0.84(±0.03) | 0.84(±0.02) | 0.82(±0.04) | 0.74(±0.03) |
| VGG16 (Imagenette) | Image | 0.80(±0.05) | 0.73(±0.03) | 0.73(±0.03) | 0.73(±0.03) | 0.70(±0.08) |
| InceptionV3 (Imagenette) | Image | 0.90(±0.05) | 0.78(±0.04) | 0.75(±0.02) | 0.74(±0.01) | 0.69(±0.04) |
| ViT_b_16 (Imagenette) | Image | 0.93(±0.01) | 0.87(±0.02) | 0.85(±0.02) | 0.78(±0.04) | 0.73(±0.08) |

Table 3: In this table, we measure the fraction of data over which the re-prediction by the complex model matches the re-prediction given by the LIPEx, upon removing Top-K features determined by LIPEx.

its top features detected be stable if only a few perturbations close to the input data are allowed. In this test, we show that not only is this true, but also that (a) LIPEx's top features can at times even remain largely invariant to reducing the number of perturbations and also that (b) the difference with respect to LIME in the list of top features detected, is maintained even when the allowed set of perturbations are increasingly constrained to be few and near the input data. Our comparison method is precisely illustrated in Algorithm C and we sketch it here as follows.

When in the setting with unrestricted perturbations, we infer two lists of top features, one from LIPEx's explanation matrix corresponding to the predicted class and another from LIME's weight vector on the same class — say LIPEx–List–s and LIME–List–s respectively. Next, we parameterize the restriction on the allowed perturbations by the maximum angle $\delta$ that any Boolean vector representing the perturbation is allowed to subtend with respect to the all-ones vector that represents the input data.

At different $\delta$, we use only the $\delta-$restricted subset of the perturbations to compute the features returned by the LIPEx and the LIME (for the predicted class), say $\delta-$LIPEx–List–s and $\delta-$LIME–List–s respectively. [9] For quantifying the dissimilarities between these lists of features calculated by the two methods, we compute the following Jaccard indices and average the results on 100 randomly chosen instances.

$$J_{s,\delta,\text{LIME}} \coloneqq \left| \frac{\delta-\text{LIME–List–s} \ \cap \ \text{LIME–List–s}}{\delta-\text{LIME–List–s} \ \cup \ \text{LIME–List–s}} \right|, \ J_{s,\delta,\text{LIPEx}} \coloneqq \left| \frac{\delta-\text{LIPEx–List–s} \ \cap \ \text{LIPEx–List–s}}{\delta-\text{LIPEx–List–s} \ \cup \ \text{LIPEx–List–s}} \right|$$

$$J_{s,\delta-\text{LIPEx–vs–LIME}} \coloneqq \left| \frac{\delta-\text{LIPEx–List–s} \ \cap \ \text{LIME–List–s}}{\delta-\text{LIPEx–List–s} \ \cup \ \text{LIME–List–s}} \right|$$

From Figure 5, we can infer that at all levels of constraint on the data at least 50% of the top features detected by our LIPEx are different from LIME. *Secondly, $J_{s,\delta,\text{LIME}}$* (averaged) rapidly falls as the number of training data allowed near the input instance is decreased. Thus its vividly revealed that the features detected by LIME are significantly influenced by those perturbations that are very far from the true text.

---

[9]In the LIME code https://github.com/marcotcr/lime, the authors choose 5000 perturbations for any text data and 1000 for any image data. And in all our LIPEx experiments we had always chosen 1000 perturbations. Hence these counts correspond to the number of perturbations chosen at $\delta = \frac{\pi}{2}$ and the count of perturbations is proportionately scales for lower $\delta$.

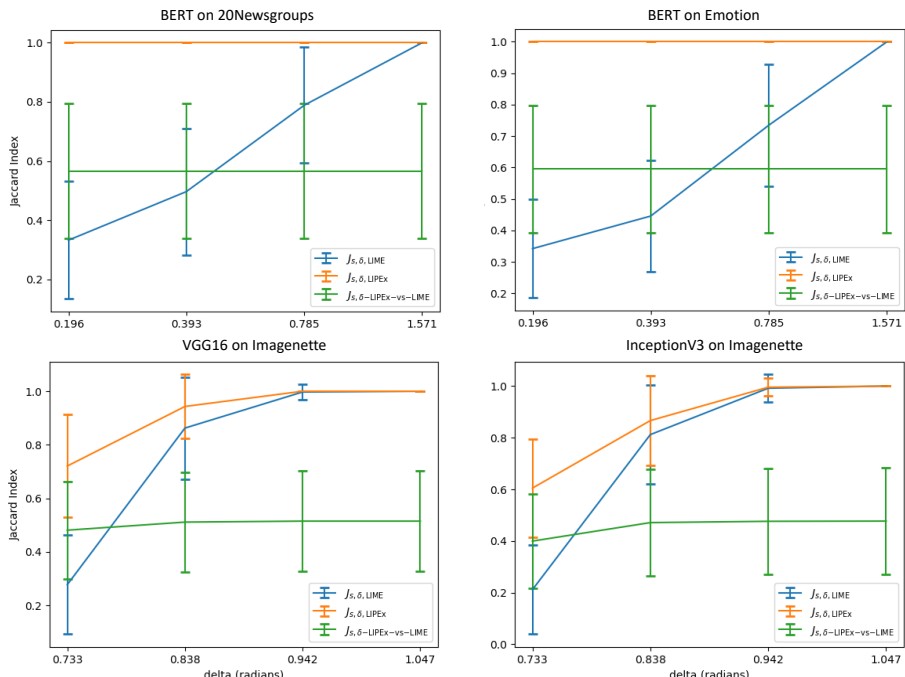

Figure 5: It can be observed that $J_{s,\delta,\text{LIPEx}}$ (the orange line) is very stable compared to that of LIME despite the allowed data perturbations being made constrained. The difference in LIPEx's features w.r.t LIME is also maintained. The number of points considered at different $\delta$ is given in Table 7 in the appendix.

*Lastly,* and most interestingly, we note that the curve for $J_{s,\delta,\text{LIPEx}}$ (the top orange line) is very stable to using only a few perturbations which subtend a low angle with the true text. Hence the top features detected by our explanation matrix are not only important (as demonstrated in the previous two experiments) – but can also be computed very data efficiently.

### 4.1 Demonstration of Computation Time Advantage of LIPEx Over LIME

In this section we take a closer look at the computation time advantage that LIPEx gets us over LIME because of its data efficiency that was pointed out above. Recall that the default setting of LIME is to use 5000 perturbations for each text data. Firstly, in Table 4 we demonstrate (over two kinds of text data) that if LIME is restricted to use only 1000 perturbations as LIPEx, then on an average there is significant degradation of the fraction of data on which prediction changes under removal of the top features detected by LIME. This demonstration motivates doing all timing tests against the default setting of LIME.

In the experiments of Table 5 it can be seen that the good setting of LIPEx and LIME using very different number of perturbations for each data shows up as a significant speed-up for LIPEx in the average time that is required to compute its matrix – as compared to the time required to get the LIME weights for each class.

Testing was done on one NVIDIA RTX 3090, and the training was done using a batch size of 128 for the Emotion dataset and 16 on the 20NewsGroups dataset, In this setting on an average LIPEx is computing its explanation matrix ~ 53% faster than all-class LIME.

## 5 Conclusion

In this work, we proposed a novel explainability framework, LIPEx, that when implemented in a classification setting, upon a single training gives a weight assignment for all the possible classes for an input with respect to a chosen set of features. Unlike other XAI methods, it is designed to locally approximate the probabilities assigned to the different classes by the complex model - and this was shown to bear out in experiments over text and images - and it withstood ablation tests. Our experiments showed that the LIPEx proposal provides

| Setting | Method | Perturbations | Top 1 | Top 2 | Top 3 | Top 4 | Top 5 |
|---|---|---|---|---|---|---|---|
| BERT (20Newsgroups) | LIME | 5000 | 0.78 | 0.84 | 0.86 | 0.88 | 0.90 |
| | | 1000 | 0.73 | 0.80 | 0.82 | 0.85 | 0.88 |
| BERT (Emotion) | LIME | 5000 | 0.64 | 0.69 | 0.71 | 0.70 | 0.75 |
| | | 1000 | 0.60 | 0.64 | 0.66 | 0.67 | 0.72 |

Table 4: Control experiments with LIME on text data shows that that for a significantly lesser proportion of data, removal of the top LIME detected features can flip the class, when the number of perturbations allowed for LIME is reduced to that of the LIPEx.

| | #Perturbations | BERT on Emotion | BERT on 20NewsGroups |
|---|---|---|---|
| LIPEx | 1000 | 8.66s | 40.82s |
| LIME | 5000 | 15.00s | 113.04s |

Table 5: Comparison of the average computational time between LIPEx and LIME on BERT over randomly chosen 100 data. Note the average text length of the Emotion dataset is 44 words while it is 441 words for the 20NewsGroups dataset.

more trustworthy explanations and it does so being more data efficient (and thus being computationally faster) than multiple other competing methods across various data modalities.

We note that our LIPEx loss, Equation 6, can be naturally generalized to other probability metrics like the KL divergence. Our studies strongly motivate novel future directions about not only exploring the relative performances between these options but also obtaining guarantees on the quality of the minima of such novel loss functions.

Also, we note that the choice of the number of top features chosen affects the quality of visualization in demonstrations like Figure 2. For text data, we treat each word as a feature, and we have a pre-defined dictionary, and a fairly sophisticated tokenizer for word splitting, which makes it relatively easier to obtain recognizable feature words. However, for image data, we cant possibly have an exhaustive dictionary of possible segments for reference. Therefore the effectiveness of the image segmentation algorithm also affects the quality of the visualization. But there are undeniable interpretability benefits in being able to explain image classification at the level of important segments, which have more semantic meaning than individual pixels. Hence it is an important direction of future research to be able to use image segmentation more robustly in conjunction with XAI methods.

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

# A  Feature Selection

For each data point, we call the "forward feature selection" algorithm below to choose the top$-k$ features for each possible class $c$. And then the feature set to be used for the columns of the LIPEx matrix is the set union of the top$-k$ features returned for each class $c$ - and we shall use $k = 5$ for our experiments.

We recall that this selection algorithm given here is a part of the LIME implementation - and we reuse it for our purposes. We also recall that in there is a code variable neighborhood_labels $\in \mathbb{R}^{\#\text{perturbations}\times\#\text{classes}}$ whose $i^{\text{th}}$ row stores the probabilities predicted (by the complex model being explained) for each class for the $i^{\text{th}}$ perturbation - which is the $i^{\text{th}}-$row of the matrix $X$ below. The vector $Y_c$ in the pseudocode given below is the $c^{\text{th}}$ column of this matrix neighborhood_labels.

---

**Algorithm 1** Forward Feature Selection for Class $c$

---

**Require:** data $X$, target $Y_c$, number of features $k$

  1:                                                                             $\triangleright\ X \in \mathbb{R}^{\#\text{Perturbations}\times\#\text{Unique}-\text{Words}}$
  2:                                                                             $\triangleright\ Y_c \in \mathbb{R}^{\#\text{Perturbations}}$
  3: `Sel_feats` $\leftarrow \{\}$
  4: $f \leftarrow$ initialize selection model of Ridge regression
  5: `All_feats` $\leftarrow \{1, 2, .., \texttt{len(X[0])}\}$                    $\triangleright$ `len(X[0])` =Number Of Unique Words in $X$
  6: **for** $i \leftarrow 1$ to $k$ **do**
  7:      `best_idx` $\leftarrow 0$
  8:      `best_score` $\leftarrow -\infty$
  9:      **for** $j \in (\texttt{All\_feats} \setminus \texttt{Sel\_feats})$ **do**
 10:          $f \leftarrow \texttt{f.fit(X[:, Sel\_feats} \cup \texttt{\{j\}], Y}_c\texttt{)}$
 11:              $\triangleright$ `f.fit()` is used to train f, where the loss function is the linear least squares with $\ell_2$-norm.
 12:          `score` $\leftarrow$ evaluate $f$ with performance metric of $R^2$
 13:          **if** `score` $>$ `best_score` **then**
 14:              `best_idx` $\leftarrow$ `j`
 15:              `best_score` $\leftarrow$ `score`
 16:          **end if**
 17:      **end for**
 18:      `Sel_feats` $\leftarrow$ `Sel_feats` $\cup \{\texttt{best\_idx}\}$
 19: **end for**
 20: **return** `Sel_feats`

---

# B  Hyperparameter Settings for all Experiments with LIPEx

A hyperparameter search was conducted over a small set of randomly selected data of each of the types mentioned below to decide on the following choices. All LIPEx experiments reported in figures 3 and 4 and tables 1, 2 and 3. were performed by using the following hyperparameter settings,

| Number of Perturbations | Learning rate | $\lambda$ | Batch size |
|:---:|:---:|:---:|:---:|
| 1000 | 0.01 | 0.001 | 128 |

Table 6: LIPEx Hyperparameter Settings

Note that $\lambda$ in above refers to the regularizer in the loss in equation 6.

## C  Pseudocode for the Quantitative Comparison Between LIPEx and LIME's Detected Important Features (as given in Section 4)

---

**Algorithm 2** LIME vs LIPEx w.r.t Angular Spread of the Perturbations About The True Data

---

**Require:** $k$ = number of top features to be used for comparing LIME and LIPEx
**Require:** A set $S$ of randomly sampled class labelled data at which the comparison is to be done
**Require:** $f^*$ = the trained predictor that needs explanations.
**Require:** $\delta-$List of all the angular deviations about the true data at which the LIPEx vs LIME comparison is to be done

1: **for** $s \in S$ **do**
2:      Compute LIPEx−List−s = top-$k$ features of $s$ w.r.t its predicted class, as detected by the LIPEx matrix using the standard set of Boolean vectors/perturbations w.r.t the all-ones representation of $s$.
3:      Compute LIME−List−s = top-$k$ features of $s$ w.r.t its predicted class, as detected by LIME using the standard set of Boolean vectors/perturbations w.r.t the all-ones representation of $s$ - on the same set of features as used in the previous step.
4:                 ▷ **Note that the above two lists of "important" features do not depend on $\delta$,**
5:                ▷ **We shall use both as reference lists for the different comparisons to follow.**
6:              ▷ **The list of features used above will be held fixed in the computations below.**
7:      **for** $\delta \in \delta-$List **do**
8:          Compute $\delta-$LIPEx−List−s = top-$k$ features of $s$ w.r.t its predicted class, as detected by the LIPEx matrix using only those Boolean vectors/perturbations which are within an angle of $\delta$ w.r.t the all-ones representation of $s$.
9:          Compute $\delta-$LIME−List−s = top-$k$ features of $s$ w.r.t its predicted class, as detected by LIME using only those Boolean vectors/perturbations which are within an angle of $\delta$ w.r.t the all-ones representation of $s$
10:
11:          Compute the Jaccard Index, $J_{s,\delta-\text{LIPEx−vs−LIME}} \coloneqq \left| \frac{\delta-\text{LIPEx−List−s} \cap \text{LIME−List−s}}{\delta-\text{LIPEx−List−s} \cup \text{LIME−List−s}} \right|$
12:          Compute the Jaccard Index, $J_{s,\delta,\text{LIME}} \coloneqq \left| \frac{\delta-\text{LIME−List−s} \cap \text{LIME−List−s}}{\delta-\text{LIME−List−s} \cup \text{LIME−List−s}} \right|$
13:          Compute the Jaccard Index, $J_{s,\delta,\text{LIPEx}} \coloneqq \left| \frac{\delta-\text{LIPEx−List−s} \cap \text{LIPEx−List−s}}{\delta-\text{LIPEx−List−s} \cup \text{LIPEx−List−s}} \right|$
14:      **end for**
15: **end for**
16: Plot $\left( \frac{1}{|S|} \cdot \sum_{s \in S} J_{s,\delta-\text{LIPEx−vs−LIME}} \right)$ vs $\delta$
17: Plot $\left( \frac{1}{|S|} \cdot \sum_{s \in S} J_{s,\delta,\text{LIME}} \right)$ vs $\delta$
18: Plot $\left( \frac{1}{|S|} \cdot \sum_{s \in S} J_{s,\delta,\text{LIPEx}} \right)$ vs $\delta$

---

## D  Additional Experiments

### D.1  Hyperparameter Choices for the Experiment in Figure 5

| For Text data | | | |
|---|---|---|---|
| $\delta$ (radians) | $\frac{\pi}{16}$ | $\frac{\pi}{8}$ | $\frac{\pi}{4}$ | $\frac{\pi}{2}$ |
| number of perturbation points | 138 | 659 | 2383 | 5000 |
| **For Image data** | | | |
| $\delta$ (radians) | $\frac{7\pi}{30}$ | $\frac{8\pi}{30}$ | $\frac{9\pi}{30}$ | $\frac{\pi}{2}$ |
| number of perturbation points | 228 | 774 | 994 | 1000 |

Table 7: The effect of $\delta$ on the number of perturbation points – averaged on 100 input instances.

Note that when $\delta$ decreases, while the amount of allowed perturbations falls, the similarity measure $\pi$ in equation 6 increases.

### D.2 Explanation Comparison Between LIPEx and LIME

In Figure 6 we consider the LIME and LIPEx explanations on a set of arbitrarily picked texts from the Emotion [10] data set. We note that this is a 6–class setup. The left-hand side matrix in the pictures is the resultant LIPEx explanation matrix (which includes its feature selection method that is built on top of LIME's as explained in Appendix A) and the right-hand side matrix is an attempted way to display the result of running LIME for each of the possible classes (with the setting that it finds the top 5 features for each class).

This side-by-side comparison vividly demonstrates that running LIME in a multi-class way can result in certain features getting picked out which do not contribute within the top-$k$ for all the classes, for a pre-fixed choice of $k$. Hence there are many missing entries - which we have marked with red cross.

This is a way to see why LIME explanations for each class cannot be stacked together into an interpretable matrix - while looking along the columns of the LIPEx matrix they can be read to give the relative importance to each class of each feature deemed to be important.

Ofcourse even if fortuitously for some data it were to be true that the LIME matrix thus formed has no missing entries, it wouldn't come with any intuition to have the key property we are after, that of being able to reproduce the distribution of the complex model being explained - the key test for LIPEx that was shown in Figures 3 and 4.

---

[10]https://huggingface.co/datasets/dair-ai/emotion

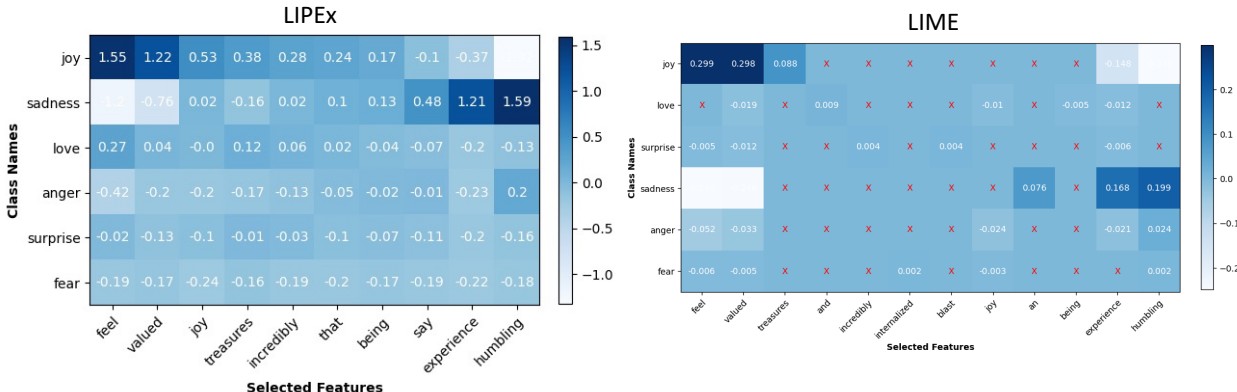

**Text:** *I have the joy of allowing kids to feel like the valued treasures that they are and to just have a blast being a kid alongside with them but can i just say its an incredibly humbling experience to have influence into a childs life and to know that what you do and say is being internalized.*

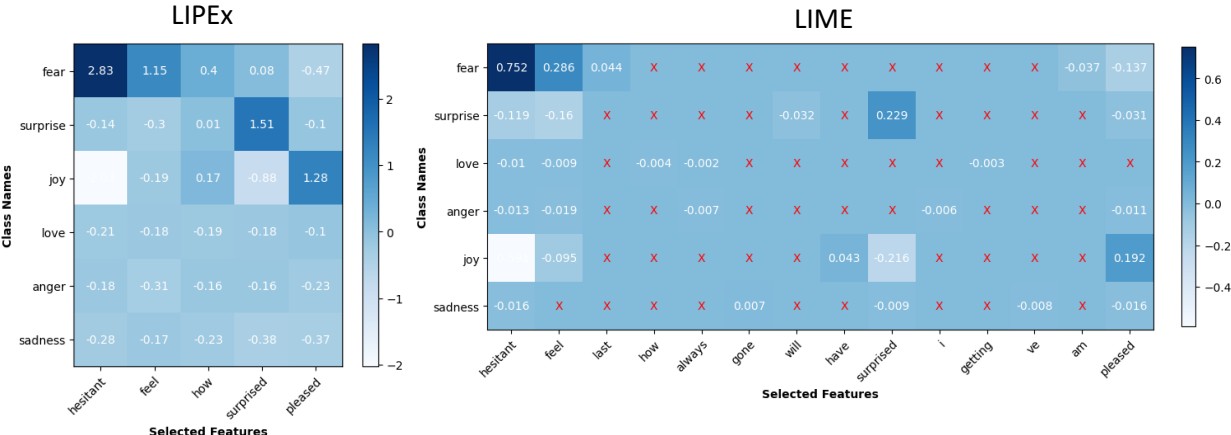

**Text:** *i feel like in the last year especially i ve gone from a girl to a woman and despite how hesitant i have always been about getting older next year i will be twenty four i am surprised at how pleased i am to have done so.*

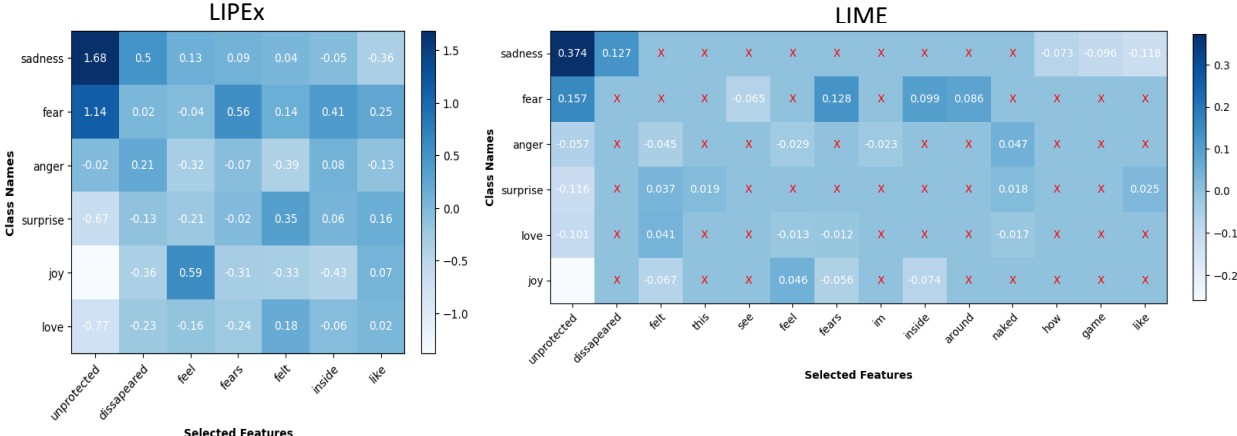

**Text:** *i feel inside this life is like a game sometimes then you came around me the walls just dissapeared nothing to surround me keep me from my fears im unprotected see how ive opened up youve made me trust coz ive never felt like this before im naked around you does it show.*

Figure 6: The above test samples are from the Emotion dataset.

### D.3 Further LIPEx vs LIME Comparisons on Longer Texts

A larger-sized version of Figure 7 can be found at https://anonymous.4open.science/r/LIPEx-7616. For longer texts we only show a comparison between the LIPEx detected matrix and the LIME weight vector that gets determined for the true/top class of the data.

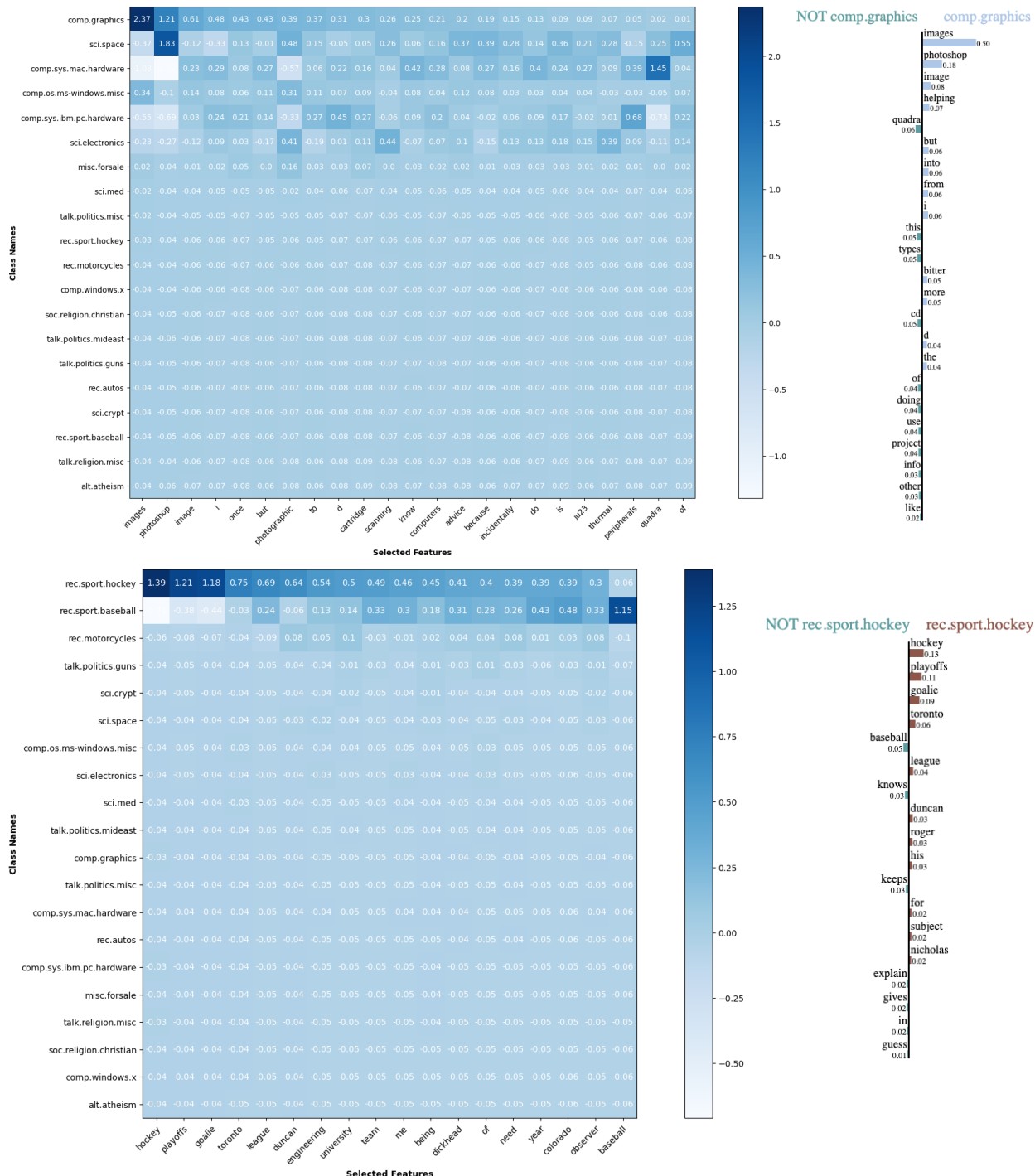

Figure 7: The above test sample is from the 20NewsGroups dataset.

## D.4 LIPEx Adheres to the Human Annotation of the HateXplain Data

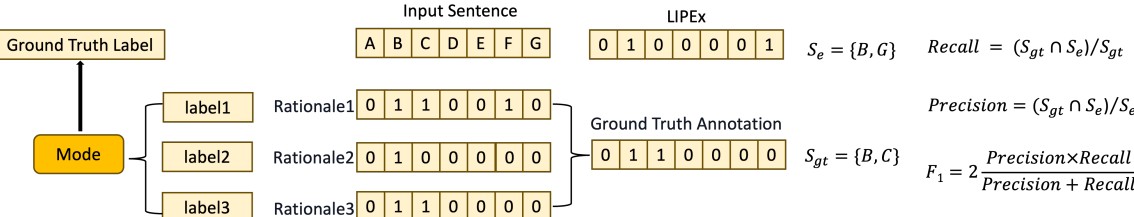

Figure 8: The diagram of dataset processing and similarity calculation between LIPEx explanation and Ground truth.

The process of this experiment is as indicated in Figure 8. The dataset we used is the HateXplain Mathew et al. (2021), an annotated dataset of Twitter and Gab posts for speech being either of the three categories hate/offensive/normal. Three annotators have labelled each post as being hateful, offensive or normal speech. Also, when a speech is determined to hateful or offensive each annotator has given rational set for that determination i.e a set of tokens/words from that sentence which they thought was being determinant to their choice. Firstly, we preprocess the dataset as below.

We filtered the dataset by removing those data points where either the rationale entry was fully empty (which corresponds to data which all three annotators thought were normal) or when each annotator gave a different label. For the remnant data we labelled it by a majority vote among the annotators and determined a ground truth annotation, say $S_{gt}$ by merging the rationales s.t those tokens were considered to be important if at least two annotators had picked that token.

Then we trained a BERT model on this processed dataset and calculated the explanation token set with LIPEx and LIME corresponding to the BERT predicted class – and this set of token determined by the XAI method is what we call $S_e$.

Finally, we filter out those data where BERT's predictions match the majority vote label given by human annotators - which is almost all of the data that BERT was trained on. We calculate Recall, Precision and F1 scores on the processed testing dataset to measure the similarity between $S_e$ and $S_{gt}$ and got the result of Recall = 0.73, Precision = 0.75, and F1 = 0.74 for LIPEx. In contrast, the result for LIME is Recall = 0.71, Precision = 0.74, and F1 = 0.72.

```
1103385226921762816_twitter:
["well", "she", "'", "muslim", "so", "of", "course", "she", "hates", "the", "jews", "jew", "hatred", "is", "one", "of", "the", "defining",
"feature", "of", "islam"]

12643541_gab:
[ "black", "people", "are", "oppressed", "by", "their", "own", "culture", "and", "community", "blacks", "need", "to", "wake", "and", "realize",
"the", "democrats", "party", "is", "the", "racist", "party", "that", "uses", "them", "for", "votes", "the", "kk", "democrats", "have", "broken",
"up", "the", "black", "family", "and", "run", "the", "community", "onto", "the", "ghetto", "plantation", "to", "control", "it", "youvebeenpunkd" ]

9551353_gab:
[ "the", "catholic", "church", "stopped", "being", "a", "bulwark", "against", "western", "rot", "when", "the", "church", "became", "infested",
"with", "homosexuals", "it", "is", "now", "just", "as", "if", "not", "more", "corrupted", "than", "the", "government", "bureaucracies", "of",
"western", "nations", "and", "has", "sunk", "so", "low", "we", "are", "hearing", "stories", "of", "depraved", "gay", "orgies", "on", "sacred",
"church", "ground" ]

18195956_gab:
[ "kick", "all", "illegal", "immigrants", "out", "they", "do", "not", "belong", "here", "they", "have", "no", "business", "being", "here", "we",
"owe", "them", "nothing" ]

13879478_gab:
[ "america", "always", "was", "a", "country", "for", "by", "and", "of", "genetic", "europeans", "a", "white", "man", "born", "in", "china", "is",
"not", "a", "chinaman", "just", "as", "a", "mexican", "anchor", "baby", "born", "in", "america", "is", "not", "american" ]
```

Figure 9: Example of lousy annotation, ID: annotation. The yellow highlighted parts are the corresponding annotations.

In Figure 9 we also point out the inherent ambiguity in the annotation of the HateXplain dataset. From the examples shown it can be seen that a lot of prepositions, pronouns and some unrelated terms are labelled as ground truth.

