# OpenReview forum: "LIPEx – Locally Interpretable Probabilistic Explanations – To Look Beyond The True Class"
_TMLR — Rejected by TMLR_

### Review · Reviewer_puXE · 2023-12-21

**Summary Of Contributions:**

The authors propose a perturbation-based multi-class explanation framework, LIPEx.
For providing information on how every feature of a text or image is important to the classes, they explain the given input in matrix format using the Hellinger distance.
The authors claim that the proposed method shows high performance in finding important features and is more efficient from the viewpoint of the computational resources used.

**Audience:**

Yes

**Broader Impact Concerns:**

It seems that there is no concern about ethical issues.

**Claims And Evidence:**

No

**Requested Changes:**

(Major) I strongly recommend that the authors assess their method on other newly developed and more complex models (i.e., ResNet50, ViT). The results of their method's performance on other models can strengthen their research. (see Weakness 1)

(Major) It would be better if the multi-class view explanation's utility and the reasons for its necessity were emphasized with other experimental settings or practical scenarios. (see Weakness 2)

(Minor) I wonder why there are no results of LIME in Figure 2.

(Minor) It is tough to understand the meaning of test 2 in Section 4. Figure 4 shows that LIPEx is more vulnerable to model distortion, so it seems to point out the weakness of LIPEx. I need an additional explanation of why it is meaningful that LIPEx tracks the changes in the complex model's output well.

(Minor) It appears that there are several typos in the document.

**Strengths And Weaknesses:**

[Strengths]
1. The authors develop a new explanation method that offers a multi-class view explanation that existing methods do not provide. Since most data (texts or images) in real-world scenarios can be related to multiple classes rather than just one, we require an explanation of the numerous classes. Thus, the motivation of this research is plausible.
2. The authors compare their method to various existing methods based on extensive experiments. Furthermore, they evaluate their method from the perspectives of performance and efficiency using a large number of ablation experiments. The authors also test that the results of their method are repeatable for different data samplings (different perturbations).
3. They suggest a trainable explanation matrix that explains complex models. They demonstrate the effectiveness of using the explanation matrix by showing how the explanation matrix's estimated probability distribution resembles one of the original models.

[Weakness]
1. The authors evaluate their method on two image classification models, VGG16 and InceptionV3. LIPEx shows relatively high performance on VGG16 compared to LIME and other methods; however, the gaps in performance between LIPEx and other methods decrease when they test on a more complex model, InceptionV3, compared to VGG16. Therefore, it is necessary to ensure that this method works well even in more complex models (i.e., ResNet50, ViT).
2. The major goal of this work is to develop the multi-class view explanation. Nevertheless, the multi-class explanation is not addressed in the section about the assessment of their method. Due to this, the answer to why the multi-class view explanation is necessary becomes ambiguous. It would be better if the multi-class view explanation's utility and the reasons for its necessity were emphasized.

---

> ### Author Response · Authors · 2024-01-26
> **Thanks for Your Comments (and We Have Updated the Draft)**
>
> We thank you for your kind reading of the draft.
> We have considered all your suggestions and submitted a revised draft.
>
> Kindly see the following clarifications for some of the specific issues you raised,
>
> >it is necessary to ensure that this method works well even in more complex models ..
>
> Kindly note that in the revision we have added our results for the ViT model too for the tests considered in Figures 3 and 4 and Tables 2 and 3. As earlier, even on this model, LIPEx continues to give better answers than its competitors.
>
> >The major goal of this work is to develop the multi-class view explanation. Nevertheless, the multi-class explanation is not addressed in the section about the assessment of their method. Due to this, the answer to why the multi-class view explanation is necessary becomes ambiguous. It would be better if the multi-class view explanation's utility and the reasons for its necessity were emphasized.
>
> Kindly note that Figures 3 and 4 is among the most critical insights and tests of LIPEx that we provide, These two figures give two different ways of establishing that LIPEx can replicate the probability distribution over classes predicted by some of the most complex neural architectures in use. Hence these two tests are direct tests of LIPEX's ability to understand the multi-class structure of predictions made by SOTA architectures in use. To the best of our knowledge, such a test is not possible with other XAI methods.
>
> Below we give further explanations on the importance of the unique test that we give in Figure 4.
>
> Separately, in the context of the multi-class explanation, our explanation matrix (as seen in say Figure 1) can concurrently identify the significant features for various classes under consideration. If we note consistent importance assigned to the same features across different classes, it suggests either the feature is not crucial or the model hasn't learned effectively. This unique insight is exclusively provided by the multi-class view, as explanations for individual classes do not offer such valuable information.
>
> > I wonder why there are no results of LIME in Figure 2
>
> We have now added the LIME results for the images considered in Figure 2, showing that for all these images, LIPEx picks out a much more informative set of segments than LIME.
>
> >  I need an additional explanation of why it is meaningful that LIPEx tracks the changes in the complex model's output well.
>
> Please note that Figure 3 has already established that for well-trained SOTA architectures, the LIPEx-determined matrix when composed with SoftMax can reproduce the predicted distribution while acting on a simple set of interpretable features. But we want to do a harder test than this - and that is what Figure 4 is after.
>
> In Figure 4 we are considering an ablation study whereby a well-trained SOTA architecture is being gradually distorted by adding larger amounts of distortions to its weights. We posit that a good explainer must always be truthful to the underlying complex model that is being attempted to be explained. So as the true model distorts, a good explainer's output distribution also needs to distort in the same way and continue to remain close to the new distributions being produced by the damaged model. And the red curve in the bottom row of figures in Figures 4, is exactly showing that  - the TV gap between LIPEx's output distribution (rederived at every value of the distorted model) and the model's output distribution always remains close to $0$ irrespective of how much distortion is done to the true model.

---

> > ### Comment · Reviewer_puXE · 2024-02-05
> > **Official comment by Reviewer puXE**
> >
> > Thank you for your answers.
> >
> > > Kindly note that in the revision we have added our results for the ViT model too for the tests considered in Figures 3 and 4 and Tables 2 and 3.
> >
> > In the revised version, I noticed that results for the ViT model have been added for the tests shown in Figures 3 and 4 and Tables 2 and 3. However, the revision shows that VGG and Inc-V3 results are missing for K=1,2. I wonder why there is no results on K=1,2 (minor issue).
> > And I still recommend that the authors evaluate their method on ResNet50 (but they presented the results on ViT, so it is just a minor issue).
> >
> > Since LIPEx shows slightly better performance compared to LIME, I believe a comparison between the two methods is important. It is recommended that the authors compare a matrix calculated by concatenating vectors obtained from LIME for all class labels, as raised by Reviewer aZvY, with W of LIPEx. This will allow for empirical validation of the authors' claim regarding the inefficiency and ineffectiveness of stacking LIME vectors.

---

> > > ### Author Response · Authors · 2024-02-07
> > > **Further Clarifications (And Draft Updated)**
> > >
> > > We thank you for your detailed reading and continued engagement with our work.
> > >
> > > Please note that in the most recently revised draft, we are showing all the tests in table-2 for the higher $K$s i.e. $3,4,5$.
> > >
> > > At very small $K$s the experiment is more noisy for certain setups and hence we drop those numbers for all for uniformity.
> > >
> > > >Since LIPEx shows slightly better performance compared to LIME, I believe a comparison between the two methods is important. It is recommended that the authors compare a matrix calculated by concatenating vectors obtained from LIME for all class labels, as raised by Reviewer aZvY, with W of LIPEx.
> > >
> > > We would like to emphasize that the most important experiment we show in favour of our method LIPEx is that in Figures 3 and 4 - which demonstrates that LIPEx can find a small set of human interpretable features over which a softmax-of-linear predictor can act and reproduce the output distribution of complex SOTA architectures in use. We posit that this observation is striking - and to the best of our knowledge existing XAI methods cannot be used to show such a similar demonstration.
> > >
> > > Additionally, it is also to be noted that there is a fundamental barrier to stacking the explanations produced by LIME into a matrix form - because it can detect non-overlapping sets of features being important for each possible class. Hence it results in "missing entries" if one were to try compiling these separate answers into a matrix - and in the updated Appendix D.2 we have given explicit examples of this phenomenon.
> > >
> > > But even if fortuitously, for some data, these missing entries were not to happen for LIME, the matrix thus obtained would still not have the guarantee to be able to reproduce the true class distribution - as in our tests in Figures 3 and 4.

---

### Review · Reviewer_KHtk · 2023-12-21

**Summary Of Contributions:**

The authors consider the task of explaining a black box classifier via local approximation with a simple explainable model as was done with LIME. They criticize that LIME attempts to regress over
bounded labels i.e. probabilities, using an unbounded function (i.e. a linear function) and that one would need to separately run LIME for each class to get the weight attributions per class label. As a remedy, the authors propose LIPEx (Locally Interpretable Probabilistic Explanations), that gives a weight assignment  for an input with respect to a chosen feature set for all the possible classes. The method is designed to locally approximate the probabilities of the black box classifier and uses a Hellinger distance loss to approximate the probability distribution of the black box classifier instead of regressing a single label output as in LIME.

The authors  experiments showed: (1) the LIPEx performs better than LIME (and various other saliency maps) on the top-K feature attribution metric; (2) LIPEx needs less perturbations than LIME and is thereby more data efficient.

**Audience:**

Yes

**Broader Impact Concerns:**

Not concerned.

**Claims And Evidence:**

No

**Requested Changes:**

On Comparisons:
To me the qualitative comparison seems flawed. The reader expects a comparison against LIME given that LIPEx makes a modification to LIME and claims superiority. Nowhere can I find that comparison. Instead the authors compare again many saliency methods that assign pixel scores and work fundamentally different. Moreover, the authors use segment anything for their method but seem to not compare to other methods with segment anything (e.g. XRAI in Figure 2 seems to use a different partition). I think there should be a comparison of LIPEx against LIME and SHAP all using the segment anything comparison. In particular, I would like to see how the LIPEx explanations look different from LIME. Is the quantitative advantage over LIME noticable in practice? This has implications on the contributions given that LIPEx introduces a relatively small modification of LIME.

On TV Gap:
The authors say "Note that the LIPEx loss (Equation 6) never directly optimized
for the TV gap to be small and hence we posit that this is a strong test of performance that LIPEx passes."
LIPEx essentially does directly optimize the TV gap since  the Hellinger distance $H$ satisfies
$H^2(P,Q) \leq \text{TV}(P,Q) \leq \sqrt{2} H(P,Q)$. So  small Hellingger distance implies small TV distance. The authors should reformulate their statement accordingly.

"Novel Explanation Framework":
The authors claim to introduce a "Novel Explanation Framework". I believe LIPEx introduces a relatively small modification to the LIME method (i.e. instead of minimizing square distance in one label output one minimizes Hellinger distance over all class label outputs).


Notation:
I would change "Soft-Max" to "SoftMax".

References:
Could the authors please use the correct reference (ICLR 2023) for
Aditya Chattopadhyay, Kwan Ho Ryan Chan, Benjamin D Haeffele, Donald Geman, and René Vidal. Vari-
ational information pursuit for interpretable predictions. arXiv preprint arXiv:2302.02876, 2023?

**Strengths And Weaknesses:**

Strenghts:
LIPEx seems to quantitatively improve upon the LIME method for classification tasks

Weakness:
- Difficult to tell how much the improvement with LIPEx over LIME is worth. In particular, the authors give qualitative example comparing explanations between LIPEx and other saliency maps on image classification but they do not compare to LIME. Is that because there is no interesting differences? Is it possible that LIME and LIPEx will qualitatively look mostly the same?
- Unsound comparisons. The reader expects also a qualitative comparison against LIME given that LIPEx makes a modification to LIME and claims superiority. Nowhere can I find that comparison. Moreover, the authors use segment anything for their method but seem to not compare to other methods with segment anything (e.g. XRAI in Figure 2 seems to use a different partition).

---

> ### Author Response · Authors · 2024-01-26
> **Thanks for the Comments (and We have Updated the Draft)**
>
> We thank you for your detailed comments.
> We have taken into account your feedback to update the submitted draft.
>
> Kindly note the following clarifications that we give to address your concerns.
>
> >Is it possible that LIME and LIPEx will qualitatively look mostly the same?
>
> We have now added the LIME results for the images considered in Figure 2, showing that for all these images, LIPEx picks out a much more informative set of segments than LIME.
>
> >The reader expects a comparison against LIME given that LIPEx makes a modification to LIME and claims superiority. Nowhere can I find that comparison. Is the quantitative advantage over LIME noticable in practice?
>
> Firstly, we would like to emphasize the question LIPEx solves i.e to reproduce the entire probability distribution of the complex model via a SoftMax-of-Linear model acting on a small set of interpretable features. This question is not even in the ambit of LIME - even when run in a multi-class way. So any comparison to LIME can be made only after we cut down LIPEx to some restricted metric like looking at the top features detected by it corresponding to the predicted class for the underlying model. It's in that sense that we show a qualitative comparison with LIME (and other XAI methods) in the few arbitrarily picked examples displayed in Figure 2
>
> We request the reviewer to also look at the quantitative metrics of comparison against LIME - as given in Tables 1,2,3,4,5 and Figure 5 which consists of data with statistical robustness.
>
> > LIPEx essentially does directly optimize the TV gap since the Hellinger distance$H$ satisfies $H^2(P, Q) <= TV(P, Q) <= \sqrt{2}H(P, Q)$. So small Hellingger distance implies small TV distance. The authors should reformulate their statement accordingly.
>
> We have now edited the language in the surrounding paragraph to make the written claim more aligned with the experiment.

---

> > ### Author Response · Authors · 2024-01-26
> > **Referencing error fixed**
> >
> > Kindly note that we have now updated the reference of the paper you pointed out to "Aditya Chattopadhyay, Kwan Ho Ryan Chan, Benjamin David Haeffele, Donald Geman, and Rene Vidal. Variational information pursuit for interpretable predictions. In The Eleventh International Conference on Learning Representations, 2022"

---

> > ### Comment · Reviewer_KHtk · 2024-02-05
> > **Reply**
> >
> > Apologies for the late reply.
> >
> > > We have now added the LIME results for the images considered in Figure 2, showing that for all these images, LIPEx picks out a much more informative set of segments than LIME.
> >
> > I am happy with this update.
> >
> > > We request the reviewer to also look at the quantitative metrics of comparison against LIME - as given in Tables 1,2,3,4,5 and Figure 5 which consists of data with statistical robustness.
> >
> > I have had another look and acknowledged before that LIPEx performs quantitatively favorably over LIME on standard feature attribution metrics.
> >
> > Beyond my previous concerns, I also agree with the point raised by Reviewer aZvY' that
> > Stacking LIME vectors and comparing the matrices would be an interesting experiment. The authors
> >
> > After a long discussion the authors responded eventually to reviewer aZvY as follows:
> > > Of course, one could use all the vocabulary as the feature set for each class and hence fill the matrix. But in trying to generate this huge matrix the code would become immensely slow - and our matrix is derivable so much faster than even the default LIME's settings. Also doing that would beat the whole point of XAI if every word possible is treated as a feature -- a core philosophy of XAI is that the underlying model should be explainable via a small number of features - as we show is possible by our way of doing feature selection and matrix finding.
> >
> > I would suggest an experiment on smaller data where the full matrix can be computed and compared. Yes, in practice the full matrix will not be computed for many datasets, however, as an experiment to analyze the difference of the methods it would be very insightful.
> >
> > To summarize remaining issues that I have:
> >
> > - I do NOT like that this work introduces LIPEx as "a novel XAI framework". It is an extension of LIME and that is okay! It has quantitative evidence and conceptual arguments that are convincing as to why approximating the full model output distribution makes more sense than a single class label. Why not frame it like that? In the end LIME is a framework and approach  that the authors build on and not something that is a distinct framework.
> > - I believe that stacking the LIME features to a matrix and analyzing the matrices (even for smaller datasets) would be an excellent and important additional experiment.
> >
> > I have a more favorable view of the work after reading the manuscript again and all the comments. However, I believe the aforementioned issues would significantly improve the work.

---

> > > ### Author Response · Authors · 2024-02-07
> > > **Further Clarifications (and Draft Updated)**
> > >
> > > We thank you for your detailed reading and continued engagement with our work.
> > >
> > > We would like to emphasize that the most important experiment we show in favour of our method LIPEx is that in Figures 3 and 4 - which demonstrates that LIPEx can find a small set of human interpretable features over which a softmax-of-linear predictor can act and reproduce the output distribution of complex SOTA architectures in use. We posit that this observation is striking - and to the best of our knowledge existing XAI methods cannot be used to show such a similar demonstration.
> > >
> > > Additionally, it is also to be noted that there is a fundamental barrier to stacking the explanations produced by LIME into a matrix form - because it can detect non-overlapping sets of features being important for each possible class. Hence it results in "missing entries" if one were to try compiling these separate answers into a matrix - and in the updated Appendix D.2 we have given explicit examples of this phenomenon.
> > >
> > > But even if fortuitously, for some data, these missing entries were not to happen for LIME, the matrix thus obtained would still not have the guarantee to be able to reproduce the true class distribution - as in our key tests in Figures 3 and 4.

---

### Review · Reviewer_aZvY · 2024-01-12

**Summary Of Contributions:**

The authors propose a new post-hoc explanation method, that is essentially a variant of LIME with the following choices:

- The surrogate model is multi-class logistic regression (instead of linear regression for each class for the standard LIME)
- The loss function is a weighted version of the Hellinger distance between the full output of the model and the surrogate's
- The regularizer is the Frobenius norm of the logistic regression weights

The authors study empirically the behavior of this new variant of LIME (called LIPEx). They show that the predictions of the surrogate are close to those of the original classifier, study the effect of adding noise or deleting features, and show that LIPEx is faster than the standard LIME, and requires less perturbations of the data.

**Audience:**

Yes

**Broader Impact Concerns:**

I have no substantial concerns.

**Claims And Evidence:**

No

**Requested Changes:**

These points are ordered by importance:

1) Doing a more thorough comparison (empirical and theoretical) with LIME, as explained in the box above.

2) Doing experiments that suggest more clearly that LIPEx's explanations are better than competitors (in particular LIME).

3) Addressing the "less pressing concerns" above

**Strengths And Weaknesses:**

# Strengths

The paper generally reads well. Even though the narration is slightly different to the one of standard ML papers (e.g. experiments are more detailed that usually in the introduction), I think it works well.

The idea of using multi-class logistic regression is very natural and sensible (and, to the best of my knowledge, novel).

While they are quite limited, the experiments are insightful.

# Weaknesses

$\textbf{1.}$ LIPEx is essentially a variant of LIME, so it would make sense to do a more thorough comparison with LIME (both in theory and practice). Here are a few things that would make the differences and similarities clearer:

The main difference is that LIPEx gives as explanation a matrix $W \in \mathbb{R}^{ \mathcal{C} \times f_x}$, where $\mathcal{C}$ is the number of classes, and $f_x$ the dimensionality of the feature space (e.g. the number of segments for an image). LIME, on the other hand, will give a vector $v_k \in \mathbb{R}^{f_x}$ for each possible class $k \in \{1,\ldots,\mathcal{C}\}$. By stacking the LIME vectors $v_1, \ldots v_\mathcal{C}$, one can obtain a matrix $V$ akin to LIPEx's $W$. Are these matrices similar ? The experiments in Figure 1, Figure 6 and Figure 7 give a partial answer, by showing that, for at least one row, this is somewhat true. If this is true more generally, what are the benefits of using LIPEx versus using LIME several time ? Clearly, it will be faster. Are there other benefits?

Since LIPEx can be seen as a variant of LIME with several specific design choices, I think that ablation studies that evaluate the individual benefits/drawbacks of these design choices would improve the paper. A few natural possible versions of LIME/LIPEx to consider in the study would be:
- LIME (potentially with the vector stacking sketched above) with the Hellinger distance instead of the standard loss to evaluate its impact
- LIPEx with the standard loss of LIME (and/or the standard L1 regularizer)

$\textbf{2.}$ I do not believe that the experiments make a convincing case that LIPEx provides better explanations than LIME. The authors experiments are essentially (interesting!) sanity checks and some convincing evidence that LIPEx is faster than LIME. It could be interesting to also look at some data sets with "ground truth explanations", like the toy tabular datasets used by Chen et al. (2019) and reused by several other authors later (Yoon et al., 2019, Senetaire et al., 2023). Senetaire et al. (2023) also introduced toy image datasets with ground truths. Agrawal et al. (2022) also introduced benchmarks.


# Less pressing concerns

- It looks like the method could be seen as a non-amortized version of Vo et al. (2023). This should be discussed.

- Some design choices are not very well motivated. Why using the Frobenius norm instead of the lasso, as in LIME ?  Why this particular weights $\pi$ ?

- I am not sure I fully agree with the premise that "it is not entirely convincing that LIME attempts to regress [...] probabilities [...] using an unbounded function (i.e. a linear function)" (p.2, seen as a motivation for the Hellinger distance). There is some theory around LIME that indicates that the problem is quite principled (e.g. Garreau and Mardaoui, 2021). Adding experimental/theoretical comparisons would help.


# Minor details

- I do not understand the sentence "there is no guarantee that by these repeated evaluations, the importance of any particular feature would be knowable for every class." (p.2)

- The authors claim that their method is unique in its way to mimic the original black-box (in p.3, "to the best of our knowledge, such a strong model replication property is not known to be true for even the saliency methods"). I think that amortized explanation methods (Chen et al., 2018, Yoon et al., 2019, Senetaire et al., 2023, Vo et al., 2023) can be able to replicate the black-box very well, since they are trained to do so.

- p.4: "Also, we recall that in works like Slack et al. (2020) it was pointed out that LIME’s reliance on perturbations far from the true data creates a vulnerability that can be exploited to create adversarial attacks." Why would it not be the case for LIPEx?

- I do not really like calling the fact that LIPEx requires less perturbations "data efficiency". Indeed, I believe that the data are what are given to the algorithm, not the artificial perturbation we create.

- I did not really understand the paragraph about  Sokol & Flach (2020), perhaps the authors can add more details.

- p.7: " Note that the LIPEx loss (Equation 6) never directly optimized for the TV gap to be small and hence we posit that this is a strong test of performance that LIPEx passes." While the TV is not directly optimized, the Hellinger is, so I do not see it as a surprise (or a "strong" test of performance, even though the experiment is definitely interesting and should stay in the paper). About this experiment, it would be interesting to also compare the accuracies of the two classifiers, this would be more interpretable than the TV.

- In Table 1 and Table 2, means and standard deviations are over 3 runs, which is not much


# References

Agarwal et al., OpenXAI: Towards a Transparent Evaluation of Model Explanations, NeurIPS 2022

Chen et al., Learning to Explain: An Information-Theoretic Perspective on Model Interpretation, ICML 2018

Garreau and Mardaoui, What Does LIME Really See in Images?, ICML 2021

Senetaire et al., Explainability as statistical inference, ICML 2023

Vo et al., An Additive Instance-Wise Approach to Multi-class Model Interpretation, ICLR 2023

Yoon et al., INVASE: Instance-wise Variable Selection using Neural Networks, ICLR 2019

---

> ### Author Response · Authors · 2024-01-26
> **Thanks for the Comments (and We have Updated the Draft)**
>
> Thanks a lot for your very detailed review of our work.
> Kindly see the updated version that we have submitted now.
>
> Our main motivation towards LIPEx is that we hope to find a set of significant features such that a linear predictor acting on them and composed with SoftMax can locally reproduce the distribution over classes that a complex classifier (like BERT) might have predicted. This in turn constitutes an explanation of the complex underlying model's predictions. To the best of our knowledge, no existing XAI method can achieve this - hence, we posit that LIPEx is achieving something fundamentally different than existing XAI approaches that we are aware of.
>
> It's a separate fact that while achieving the above, for various text data, LIPEx also happens to be faster at getting multi-class explanations than calling LIME separately on each class.
>
> In what follows we will clarify the key points that you have raised,
>
> > A few natural possible versions of LIME/LIPEx to consider in the study would be: LIME (potentially with the vector stacking sketched) with the Hellinger distance instead of the standard loss to evaluate its impact..
>
> We would like to point out that Hellinger distance makes sense only when comparing two probability distributions. LIME's predictor is an unbounded linear function and hence it's not a valid idea to compare it with a distribution.
>
> Also, LIME's mechanism does not ensure that for each class the importance would be determined over the same set of features. Hence there is no natural way to stack the LIME explanations for the different classes into a matrix form. LIPEx is designed to find an explanation that maps a set of features to the distribution over classes predicted by the complex classifier.
>
> > It could be interesting to also look at some data sets with "ground truth explanations".
>
> We have added a supplementary experiment with the HateXplain dataset with ground truth in Appendix D.3. We have carefully set up the experiment to show that LIPEx not only has a significant overlap with the human annotations when they exist but that it also gets slightly better results than LIME on that metric.
>
> > It looks like the method could be seen as a non-amortized version of Vo et al. (2023)
>
> The study in https://openreview.net/pdf?id=5OygDd-4Eeh also produces an explanation for each class at the same time in a multi-class set-up. However, there seems to be a big and fundamental difference with our work - that our method aims to and successfully reproduces the probability distribution over classes as the original model, whereas in their work neither the formalism as stated nor any of their given experiments can be interpreted to have this critical ability that drives our work.
>
> We posit that almost all standard multi-class classification experiments are designed so that the complex model gives a distribution over classes for the data and hence a good XAI method must explain/reproduce this entire distribution - which is what we achieve in this work.
>
> > Why using the Frobenius norm instead of the lasso, as in LIME ? Why this particular weights $\pi$?
>
> We recall that weight-decay is a very standard regularization method in deep-learning and that comes from having a Frobenius norm regularization. Thus we were inspired to use that as our first choice for regularizing the LIPEx matrix. Regularlizing by the $\ell_1-$norm would help promote sparsity of the explanation and its influence on the quality of the explanations can be a later study - the scope of this paper is to establish that probabilities predicted by complex models can at all be explained.
>
>  We note that the weighting of $\pi$ that we do over the perturbations is something that already existed in LIME and we clearly see the benefits of continuing with that idea.
>
> >I do not understand the sentence "there is no guarantee that by these repeated evaluations, the importance of any particular feature would be knowable for every class." (p.2)
>
> We recall that to use LIME in a multi-class way the LIME algorithm needs to be rerun from scratch for every class possible for the underlying model. And LIME's mechanism does not ensure that for each class the importance would be determined over the same set of features. Hence there is no natural way to stack the LIME explanations for the different classes into a matrix form. But LIPEx is designed to find an explanation that maps a set of features to the distribution over classes predicted by the complex classifier.

---

> ### Author Response · Authors · 2024-01-26
> **Contd.**
>
> >p.4: "Also, we recall that in works like Slack et al. (2020) it was pointed out that LIME’s reliance on perturbations far from the true data creates a vulnerability that can be exploited to create adversarial attacks." Why would it not be the case for LIPEx?
>
> Compared with LIME, LIPEx can work with a small set of perturbations all chosen close to the true data without compromising accuracy (as shown in Figure 5), and hence that gives the intuition as to why LIPEx might be less vulnerable to an adversarial strategy that depends on the XAI method using perturbations far from truth. We posit that this proof of the robustness of LIPEx is a potentially interesting direction for future research.
>
> >I do not really like calling the fact that LIPEx requires less perturbations "data efficiency". Indeed, I believe that the data are what are given to the algorithm, not the artificial perturbation we create.
>
> To get the explanation matrix the only data that is needed are the perturbations of the true data at which the underlying complex model is being explained (as seen in the loss function given explicitly in equation 6). It's in this sense that we use the phrase "data efficiency".
>
> >While the TV is not directly optimized, the Hellinger is, so I do not see it as a surprise (or a "strong" test of performance, even though the experiment is definitely interesting and should stay in the paper).
>
> We have now edited the language in the surrounding paragraph to make the written claim more aligned with the experiment.
>
> >In Table 1 and Table 2, means and standard deviations are over 3 runs, which is not much
>
> Kindly note that in any one run, the XAI results have to be computed for all the methods considered for randomly chosen $100$ data points. This results in a huge computation requirement to generate these tables 1,2 and 3 - and this is already at a stretching point for our limited resources.

---

> > ### Comment · Reviewer_aZvY · 2024-01-29
> >
> > Hi,
> >
> > Many thanks for your detailed answer. There are a few things that I do not fully understand still.
> >
> > > We would like to point out that Hellinger distance makes sense only when comparing two probability distributions. LIME's predictor is an unbounded linear function and hence it's not a valid idea to compare it with a distribution.
> >
> > For binary classification, the unbounded function is just the estimated log-odds of class $1$ against class $0$, which can be used to compute any proper scoring rule, making it easy to compare it to a distribution. The original LIME paper considers a generic loss function $\mathcal{L}$, which could be chosen to be the Hellinger distance.
> >
> >
> > > LIME's mechanism does not ensure that for each class the importance would be determined over the same set of features. Hence there is no natural way to stack the LIME explanations for the different classes into a matrix form.
> >
> > Say we deal with images. LIME's set of features is given by running a segmentation algorithm, which does not depend on the class. So why would the feature set vary?
> >
> > > there seems to be a big and fundamental difference with our work - that our method aims to and successfully reproduces the probability distribution over classes as the original model, whereas in their work neither the formalism as stated nor any of their given experiments can be interpreted to have this critical ability that drives our work.
> >
> > It seems that Vo et al. (2023) minimise the cross entropy between the black-box model and the explainable one, in a somewhat similar fashion as you optimise the Hellinger. Therefore, I think it is also trained to reproduce the probabilistic predictions of the original model.

---

> ### Author Response · Authors · 2024-02-01
> **Further Clarifications**
>
> Thanks a lot for your comments. We address them as follows,
>
> >The original LIME paper considers a generic loss function $\mathcal{L}$, which could be chosen to be the Hellinger distance.
>
> Inside the LIME setup, we could always replace the underlying predictor by a transformation of its unbounded output into a probability and then regress against that. But that would entail changing the explainer's structure to also give probabilities as output - and then we would essentially be back to a variant of LIPEx restricted to binary classification!
>
> We posit that if we follow this style of reasoning as you proposed to its logical end and to fully multi-class settings, one would naturally be led to the LIPEx formulation - except that the goodness of our choice of the probability metric (i.e Hellinger) is empirical and our conclusion already points out future directions of being able to decide its merits on theoretical grounds.
>
> We posit that LIPEx's driving philosophy of casting XAI as a regression problem in the probability space is a very natural way to do XAI which subsumes many such intuitions and we show that the resultant method outperforms on benchmark tests across multiple data modalities as evidenced in the submission.
>
> >Say we deal with images. LIME's set of features is given by running a segmentation algorithm, which does not depend on the class. So why would the feature set vary?
>
> Firstly we recall that the image segmentation algorithm is run independent of any future XAI use of the produced segments for the image. Next, we realize that LIME has a feature selection stage that will select a feature subset for each specific class - and this feature set to be used for explaining can depend on the target class to be explained. This can be seen from studying line 183 of the LIME source code (https://github.com/marcotcr/lime/blob/fd7eb2e6f760619c29fca0187c07b82157601b32/lime/lime_base.py#L182-L187) which calls the feature selection function to choose the important features for the corresponding class (corresponding to the label parameter at line 141).
>
> A more vivid example can be found in this (multi-class demonstration of LIME) (https://marcotcr.github.io/lime/tutorials/Lime%20-%20multiclass.html) -- where it can be seen that the two bar charts output on line [17] (and also compare to line [14])  correspond to the same piece of text but the selected features for deciding the relative weights are different depending on which class is being explained.
>
> The fact that LIME chooses a different set of features for different classes is a barrier to any immediate recasting of that method to an explainer of the full information content of the underlying model i.e the predicted distribution.
>
> >It seems that Vo et al. (2023) minimise the cross entropy between the black-box model and the explainable one, in a somewhat similar fashion as you optimise the Hellinger. Therefore, I think it is also trained to reproduce the probabilistic predictions of the original model.
>
> We would like to emphasize that our similarity with Vo et. al. (2023) (https://openreview.net/pdf?id=5OygDd-4Eeh) is only up to the fact that both the methods give as output a matrix of the same shape. Their equation 5 indicates that multiple neural nets need to be trained to find this explanation matrix - while in contrast, we are just doing regression in a much smaller dimensional space and extracting entire distributional information.
>
> In Vo et. al. (2023) the quantity $\tilde{y}_m$ that occurs in $\mathcal L_1$ and $y_m$ that occurs in $\mathcal L_2$ are class labels and not any kind of probabilities predicted by the underlying model. So the XAI method they have proposed is only targeted to reproduce the top class predicted by the underlying model and the explainer never utilizes the information that a distribution over classes was produced by the underlying model - as is typical of most multi-class classification experiments.  This is also reflected in the metrics they check for as listed in their Appendix B. To the best of our understanding, they do not claim to be able to reproduce the full distribution.
>
> But our XAI method is targeted to solve this very question - that of being able to reproduce in an interpretable way the entire set of probabilities over classes that was predicted. Thus we obtain an explanation for not just the top class but also for all the other possibilities over a fixed set of features and thus leading to a richer view allowing for direct comparison of contributions of features across classes. Our tests given in Figures 3 and 4 are the core evidence of our method's ability - and it brings to light a very surprising fact that complex SOTA architectures produce distributions that are locally easily replicable over human interpretable features.

---

> > ### Comment · Reviewer_aZvY · 2024-02-02
> >
> > Many thanks for your response.
> >
> > > But that would entail changing the explainer's structure to also give probabilities as output - and then we would essentially be back to a variant of LIPEx restricted to binary classification!
> > > We posit that if we follow this style of reasoning as you proposed to its logical end and to fully multi-class settings, one would naturally be led to the LIPEx formulation
> >
> > I essentially agree, this is why I mostly see LIPEx as a multi-class extension of LIME.
> >
> > >A more vivid example can be found in this (multi-class demonstration of LIME) (https://marcotcr.github.io/lime/tutorials/Lime%20-%20multiclass.html) -- where it can be seen that the two bar charts output on line [17] (and also compare to line [14]) correspond to the same piece of text but the selected features for deciding the relative weights are different depending on which class is being explained.
> > > The fact that LIME chooses a different set of features for different classes is a barrier to any immediate recasting of that method to an explainer of the full information content of the underlying model i.e the predicted distribution.
> >
> > Here, I still do not follow. In LIME, the explanation of each class is a sparse vector. These sparse vectors have indeed different sparsity patterns, but nothing is preventing us from stacking them still. For instance, using the LIME tutorial notebook that you cite, I ran the following code to generate a matrix similar to yours, where each row corresponds to a class. This is quite inefficient code but it seems to work (e.g. using a sparse matrix would be much better).
> >
> > ```
> > exp = explainer.explain_instance(newsgroups_test.data[idx], c.predict_proba, num_features=6, labels = range(0,len(explainer.class_names))) # to get the explainations of all classes
> >
> > vocabulary = vectorizer.vocabulary_ # the whole text vocabulary
> >
> > exp_matrix = np.zeros(shape = [ len(vocabulary),  len(explainer.class_names)])
> >
> > for label in range(0,len(explainer.class_names)):
> >
> > # we fill each column of the matrix sequentiallty
> >
> >   label_exp = exp.as_list(label)
> >
> >   for feature, weight in label_exp:
> >
> >     # for a given colum, we fill the corresponding row
> >
> >     exp_matrix[vocabulary[feature],label] = weight
> >
> > ```
> >
> > > Their equation 5 indicates that multiple neural nets need to be trained to find this explanation matrix - while in contrast, we are just doing regression in a much smaller dimensional space and extracting entire distributional information.
> >
> > It is true that amortized methods train multiple neural nets to produce explanations, which is a drawback. On the other hand, they have only to be trained once for the whole dataset, it is not required to do it for every instance.
> >
> >
> > > So the XAI method they have proposed is only targeted to reproduce the top class predicted by the underlying model and the explainer never utilizes the information that a distribution over classes.
> >
> > Indeed! I misunderstood their paper and thought they were sampling from the softmax, but they just take the argmax is seems. Sorry for the misunderstanding. I believe the other papers on amortized explaination I mentioned (Chen et al., 2018, Yoon et al., 2019, Senetaire et al., 2023) do sample from the predictive density though, and are trained to mimic this predictive density.

---

> ### Author Response · Authors · 2024-02-05
> **Further Replies**
>
> We thank you for your careful reading and your detailed comments and dedicated engagement with our work.
>
> We would like to posit that because of our ability to reproduce the distribution of the underlying model, LIPEx goes significantly beyond what the authors of LIME called as multi-class LIME - for which we always refer to the official multi-class demonstration that was provided  https://marcotcr.github.io/lime/tutorials/Lime%20-%20multiclass.html
>
> Our section 4.1 already gives an example of timing comparison to show how significantly faster LIPEx can be than the above way of using LIME.
>
> This is a tell-tale sign that the change between the two methods is much fundamental
>
> -- and we posit that it arises because we frame the XAI question in the distribution space.
>
> > This is quite inefficient code but it seems to work
>
> We would like to point out that the matrix your code snippet is constructing does not address the issue we raised as one of our motivations for our invention. Firstly, the setting of "num_features=6" means that for each possible class you will only be returned a weight vector over its top-6 relevant features. This means that most of the entries in this matrix will not be filled (given that it is using every word in the vocabulary as a potential feature). (This is not a result of a sparse matrix regression either but rather missing values in the purported explanation matrix.) This is what we meant when saying that there is no natural way to stack LIME outputs for every class into a matrix like what we derive.
>
> Ofcourse, one could use all the vocabulary as the feature set for each class and hence fill the matrix. But in trying to generate this huge matrix the code would become immensely slow - and our matrix is derivable so much faster than even the default LIME's settings. Also doing that would beat the whole point of XAI if every word possible is treated as a feature -- a core philosophy of XAI is that the underlying model should be explainable via a small number of features - as we show is possible by our way of doing feature selection and matrix finding.
>
> The surprise in our method is that it reveals that a matrix over a small set of features can be used to get the distribution so correctly for SOTA models across vision and text tasks!
>
> Lastly, even if this all-vocabulary matrix is computed using an enormous amount of time, it still wont automatically suffice to replicate the distributions of the target complex model
>
> -- which is possible by LIPEx and we have vividly displayed that in the tests shown in Figures 3 and 4.
>
> > (Chen et al., 2018, Yoon et al., 2019, Senetaire et al., 2023) do sample from the predictive density though...
>
> We would like to emphasize that methods L2X, INVASE and LEX that is referred to here are not post-hoc explaination methods like LIME or LIPEx is. None of these three ideas are built to explain the predictions of an arbitrary given model but each proposes its own method to design a classifier which is more interpretable than the standard models - and they all share the feature of training multiple nets to achieve it.
>
> So the question doesn't even arise of them being sensitive or not to the distribution output at any input to a target complicated model that needs to be explained. There is no natural way that one can even define for them the critical distribution replication tests that we show in Figures 3 and 4.

---

> > ### Comment · Reviewer_aZvY · 2024-02-07
> >
> > Thanks for your response.
> >
> > I agree that the speed difference is valuable (although the speed bottleneck is generally to train the predictor, not the explainer), but I think the paper would be more interesting if a comparison with simple baselines such as "stacking LIME" were conducted.
> >
> > >  This means that most of the entries in this matrix will not be filled (given that it is using every word in the vocabulary as a potential feature). (This is not a result of a sparse matrix regression either but rather missing values in the purported explanation matrix.)
> >
> > I don't understand. It is my understanding that these entries are zeroes, and not missing, precisely because of sparse regression used in standard LIME. If you were to use LIME with ridge regression instead of sparse regression, you would get a dense matrix. Here, using sparse regression brings about a sparse matrix.
> >
> >
> > > We would like to emphasize that methods L2X, INVASE and LEX that is referred to here are not post-hoc explaination methods like LIME or LIPEx is. None of these three ideas are built to explain the predictions of an arbitrary given model but each proposes its own method to design a classifier which is more interpretable than the standard models - and they all share the feature of training multiple nets to achieve it.
> >
> > This is not the case. The starting point of L2X is to explain a given model, denoted $\mathbb{P}_m( y | x)$ in the L2X paper. As discussed in Section 2.5 of the LEX paper, it seems that these methods can both do post-doc explanations, and or an interpretable classifier, depending on how people use them.

---

> ### Author Response · Authors · 2024-02-07
> **Clarifications**
>
> Thanks a lot for your comments and continued engagement with our work.
>
> *Firstly,* we would like to clarify that the time required to train the underlying model (like BERT) is not a part of the timing measurements that we have reported in Section 4.1. The complex model to be explained is considered as being given. In there, we always report the time required to approximately solve the required optimizations to get the explanation weights -- for LIME as well as LIPEx. So the advantage that LIPEx shows there comes entirely from the difference it has w.r.t LIME in how its loss function (equation 6) is structured. More specifically we believe that the advantage largely comes because LIPEx finds the multi-class explanation using a far lesser number of perturbations than LIME - as explained in Section 4.1.
>
>
> *Secondly,*  we would like to draw your attention to the updated Appendix D.2 where for certain texts we have given vivid side-by-side comparisons between the LIPEx explanation matrix and an attempt to compile into a matrix the multi-class LIME weights (of top-5) features for each class. The figure in Appendix D.2 explicitly shows the "missing entry" phenomenon that we referred to - that once the threshold value of k is fixed, LIME can find non-overlapping sets of top-k features for each class and hence it's very easily possible that when trying to compile such a matrix, for certain features their contributions to every class won't get determined. These missing entries are what we have marked with a red cross in the demonstration there.
>
> One can also look into lines 183 to 193 of the LIME code,
> https://github.com/marcotcr/lime/blob/fd7eb2e6f760619c29fca0187c07b82157601b32/lime/lime_base.py and see that the feature selection process initially picks a subset of features, after which LIME applies the regression function to this subset. Therefore the importance cannot be determined for the features not included in the regression - and this has nothing to do with what kind of penalty (sparsity or not) is used in the regression therein.
>
> But even if fortuitously, for some data, these missing entries were not to happen for LIME, the matrix thus obtained would still not have any reason to be able to reproduce the true class distribution - as shown in our crucial distributional tests for LIPEx in Figures 3 and 4.

---

> > ### Comment · Reviewer_aZvY · 2024-02-11
> >
> > I fully agree with your first point, but do not agree with the second one.
> >
> > > One can also look into lines 183 to 193 of the LIME code, https://github.com/marcotcr/lime/blob/fd7eb2e6f760619c29fca0187c07b82157601b32/lime/lime_base.py and see that the feature selection process initially picks a subset of features, after which LIME applies the regression function to this subset. Therefore the importance cannot be determined for the features not included in the regression - and this has nothing to do with what kind of penalty (sparsity or not) is used in the regression therein.
> >
> > It is very common in sparse regression to initially select the features, then perform regression on these features. This is a quite standard way to do sparse regression (see, e.g. Belloni and Chernozhukov, 2013). The features zeroed in the first step are not missing, in my opinion, but truly discarded. You could replace this by just doing ridge regression and get a full matrix.
> >
> > Belloni and Chernozhukov, Least squares after model selection in high-dimensional sparse models, Bernoulli, 2013

---

> ### Author Response · Authors · 2024-02-12
> **Clarifications**
>
> Thanks for your reply.
>
> We agree with the essential spirit of your comment "The features zeroed in the first step are not missing, in my opinion, but truly discarded."
>
> We were using the phrase "missing" to describe the same thing as what you called "discarded"
>
> -- and this is what results in the red-cross marks appearing in our demonstration in Appendix D.2
>
> The key point is that, because LIME in trying to get a parsimonious explanation will discard various features for each class (without any communication across classes) it will result in getting incomplete matrices -- where for certain features we won't know what is their contribution to every class. And, hence multi-class LIME explanations cannot be stacked together reasonably.
>
> Ofcourse, we can brute-force out of this limbo by using all the vocabulary as the feature set for each class and hence fill the matrix. But in trying to generate this huge (O(hundred) to O(thousand) dimensional for any usual dataset!) matrix the LIME code would become even slower than already,
>
> -- and that would then beat the whole point of XAI if every word possible is treated as a feature.
>
> We posit that a core philosophy of XAI is that the underlying model should be explainable via a small number of features
>
> -- as we show is possible by our way of doing feature selection and matrix finding.
>
> Even if fortuitously, for some data, these missing entries were not to happen for LIME, even for its own way of choosing top-few features for each class --  the matrix thus obtained would still not have any reason to be able to reproduce the true class distribution - as shown in our crucial tests for LIPEx in Figures 3 and 4. The surprise in our method is that it reveals that a matrix over a small set of features can be used to get the distribution so correctly for SOTA models across vision and text tasks!

---

### Decision · Action_Editor_nAy5 · 2024-02-18

**Recommendation:** Reject

**Comment:**

I believe that this paper is generally well-written, and the contribution is not trivial. However, this paper overclaims their contribution in terms of novelty. Note that novelty is not the main criterion of TMLR, but overclaiming novelty without sufficient evidence can be problematic. After the long discussions between the authors and the reviewers, I think the paper's novelty is somewhat overclaimed, and the evidence to support the novelty is insufficient.

As all the reviewers agreed, this paper needs to tone down its novelty and contain more fair experimental results with LIME. I generally agree with the Reviewer KHtk's final comment. Please revise the paper following the comment. I think if the overclaim part is resolved, some audiences will be interested in this paper.

As a minor comment, I believe that this paper could be more concise to contain all the important content in the Appendix. At least, I believe that D.2~4 should be contained in the main paper (maybe excluding verbose tables). Please consider shortening the main manuscript and re-organizing appendix texts to the main paper. If the authors think it is not possible to reduce the main manuscript, please consider submitting the paper as a long paper (more than 12 pages) to make the overall text clearer.

**Audience:**

Explainable AI (XAI) is a widely studied field, and extending the popular LIME algorithm would be interesting for some audiences. However, this paper will need more fair evaluation with the naive extension of LIME.

**Claims And Evidence:**

This paper introduces LIPEx (Locally Interpretable Probabilistic Explanation), a perturbation-based multi-class local explanation framework. Compared to LIME, LIPEx is able to be applied to the multi-class classifiers by employing multi-class logistic regression, while LIME uses a single-class regression. While the authors argue that LIPEx is a novel framework, all the reviewers agree that the proposed method is highly similar to LIME. In my opinion, it is okay to propose a similar or extended version of the existing algorithm (especially for TMLR), but as the reviewers agreed, the argument for the method novelty in this paper is somewhat overclaimed.

I believe that some arguments regarding LIME require more evidence. For example, the authors claimed that the feature importance of some features is "missing" for the standard LIME, but as Reviewer aZvY's comment, I think this argument needs more evidence.

Also, while the authors claimed that LIPEx cannot be directly compared to LIME during the response period, as all the reviewers agreed, it could be simply done by stacking the LIME results to get a matrix (see Reviewer aZvY's comment for details). Without the suggested experiment by reviewers, it is hard to argue that LIPEx is more data efficient (using a smaller number of perturbations to obtain a reliable explanation) than LIME, as the paper claims. Furthermore, as Reviewer puXE pointed out, Table 2 does not include K=1 and 2 results, whereas Table 1 reports K=1-5 and also contrasts with the initial inclusion of K=1-5 in Table 2. As Reviewer puXE's argument, I also believe that these results could lead to doubt of the reliability of the evaluation.

Overall, I don't think this paper's claim is sufficiently supported by accurate, convincing and clear evidence.

**Resubmission Of Major Revision:**

The authors may consider submitting a major revision at a later time.